DOI: 10.1038/s41467-018-06620-4　　**OPEN**

# Large-scale whole-exome sequencing association studies identify rare functional variants influencing serum urate levels

Adrienne Tin ⬤ et al.[#]

Elevated serum urate levels can cause gout, an excruciating disease with suboptimal treatment. Previous GWAS identified common variants with modest effects on serum urate. Here we report large-scale whole-exome sequencing association studies of serum urate and kidney function among ≤19,517 European ancestry and African-American individuals. We identify aggregate associations of low-frequency damaging variants in the urate transporters *SLC22A12* (URAT1; $p = 1.3 \times 10^{-56}$) and *SLC2A9* ($p = 4.5 \times 10^{-7}$). Gout risk in rare *SLC22A12* variant carriers is halved (OR = 0.5, $p = 4.9 \times 10^{-3}$). Selected rare variants in *SLC22A12* are validated in transport studies, confirming three as loss-of-function (R325W, R405C, and T467M) and illustrating the therapeutic potential of the new URAT1-blocker lesinurad. In *SLC2A9*, mapping of rare variants of large effects onto the predicted protein structure reveals new residues that may affect urate binding. These findings provide new insights into the genetic architecture of serum urate, and highlight molecular targets in *SLC22A12* and *SLC2A9* for lowering serum urate and preventing gout.

Correspondence and requests for materials should be addressed to A.T. (email: atin1@jhu.edu) or to O.M.W. (email: owoodward@som.umaryland.edu) or to A.K. (email: anna.koettgen@uniklinik-freiburg.de). [#]A full list of authors and their affliations appears at the end of the paper.

The kidney has a fundamental role in whole body homeostasis[1]. It is essential in filtering blood and actively reabsorbing and secreting solutes, metabolites and proteins during tubular passage. The importance of tubular kidney function is illustrated by the regulation of urate metabolism. Serum urate is freely filtered by the kidney followed by a complex balance of tubular reabsorption and secretion[2]. Many genes that participate in the tubular reabsorption of urate are primarily expressed in the kidney[2]. Elevated serum urate levels, hyperuricemia, can cause gout, an excruciating disease that affects 4% of the U.S. population and up to 10% of high risk populations[3,4]. The treatment of gout is suboptimal resulting in recurrent gout flares[5]. Our understanding of human physiology and pathophysiology of both glomerular and tubular kidney function is still limited. Genetic studies offer the possibility to gain novel insights into the molecular mechanisms of these kidney-related traits.

The genetic predisposition to hyperuricemia and kidney dysfunction is evidenced by monogenic diseases and population-based studies[6,7]. Studies of individuals of European ancestry (EA), serum urate, estimated glomerular filtration rate (eGFR), a measure of kidney function, and urinary albumin-to-creatinine ratio (UACR), a measure of kidney damage, have reported heritabilities of 40–70, 33, and 16%, respectively[8-10]. Genome-wide association studies (GWAS) have identified common variants for these traits that can account for 8, 5, and 1% of the serum urate, eGFR, and UACR variances, respectively[11-13]. However, the effect size of each common variant was modest, and few causal GWAS variants were supported through experimental evidence, except for the common missense variant Q141K in ABCG2, which encodes a urate transporter[14]. Evidence from monogenic diseases and from large-scale association studies[15] support a negative correlation between variant frequency and effect size, suggesting that the systematic study of low-frequency variants can identify variants of larger effect.

Therefore, we aim to identify low-frequency (minor allele frequency [MAF] 1–5%) or rare (MAF < 1%) variants associated with serum urate, eGFR, and UACR through whole-exome sequencing (WES) association studies in up to 19,517 EA and African-American (AA) individuals, and to translate genetic associations into mechanistic insights. Here, we identify novel associations between serum urate and low-frequency damaging variants in the urate transporters SLC22A12 and SLC2A9, experimentally validate loss-of-function variants in SLC22A12, and identify candidate residues important for urate binding in SLC2A9. These findings provide new insights into the genetic architecture of serum urate and gout, and highlight attractive molecular targets in SLC22A12 and SLC2A9 for lowering of serum urate and prevention of gout.

## Results

**Study populations and genomic control of meta-analyses**. This study included up to 19,517 participants (15,821 EA and 3,696 AA) from eight cohorts. The total analyzed sample sizes were 16,189 for serum urate, 6762 for gout, 18,795 for eGFR, and 12,844 for UACR. The mean age in each cohort ranged from 53 to 78 years. The population characteristics of each cohort are reported in Supplementary Data 1. We analyzed the association of 781,062 variants with minor allele count (MAC) ≥ 10 identified through WES that mapped into 19,549 genes. Genomic control factors for the primary single-variant meta-analyses ranged from 0.99 to 1.00 and 1.05 to 1.07 for the primary optimal sequence kernel association test (SKAT-O) meta-analyses.

**Summary of the studies of kidney function**. The studies of the kidney function measures, eGFR and UACR, confirmed several known associations, including the burden of rare variants in SLC47A1 for eGFR[16]. Details of post-hoc power analyses, as well as the eGFR and UACR results are reported in Supplementary Note 1 and Supplementary Data 2 to 8.

**Results of single-variant meta-analysis of serum urate**. The overall single-variant meta-analysis of serum urate levels identified 16 variants as exome-wide significant ($p < 7.8 \times 10^{-8}$, Supplementary Data 9). Of these, 13 variants were common (MAF ≥ 5%) and located in known serum urate GWAS loci, including rs2231142 (Q141K) in ABCG2 (beta: 0.22, $p = 2.4 \times 10^{-32}$; OR for gout: 2.05, $p = 1.7 \times 10^{-15}$), and other variants in SLC2A9, ABCG2, GCKR, SLC17A1, and SLC17A4[14]. For the eight exome-wide significant common variants in the SLC2A9 region, analyses controlling for the GWAS index variant, rs12498742[13], among 6545 EA participants of the ARIC study showed effect size attenuations for all eight variants that ranged from 40 to 67% (Supplementary Data 10).

**Low-frequency coding variants in SLC22A12 and SLC2A9**. Two of the exome-wide significant variants were rare, independent missense variants in SLC22A12 with relatively large effects: rs150255373 (beta = −1.11 SD [~−1.67 mg/dL], $p = 1.3 \times 10^{-10}$) and rs147647315 (beta = −0.74 SD [~−1.11 mg/dL], $p = 1.1 \times 10^{-10}$, Table 1). The variant rs150255373 was observed exclusively in EA participants, while rs147647315 was observed mainly among AA participants (70 of 77 copies) and has been reported in the Human Gene Mutation Database (HGMB) as linked to hypouricemia (accession number: CM1110340)[17]. The novel association of rs150255373 with serum urate was independent of the reported GWAS index variant, rs478607[13], as demonstrated in conditional analysis including 6713 EA participants of the ARIC study (Supplementary Data 10). The secondary analysis on the

---

**Table 1 Novel rare missense variants in SLC22A12 associated with serum urate in the primary single-variant meta-analysis combining EA and AA individuals**

| Variant | Amino acid position[a] | Alt/ref allele | Serum urate | | | | | Gout | | | | | |
| | | | Alt. allele count | Beta[b] | SE | P-value | N | Alt allele count | OR | Beta | SE | P-value | N |
|---|---|---|---|---|---|---|---|---|---|---|---|---|---|
| rs150255373 | R325W | T/C | 34 | −1.11 | 0.17 | 1.3E-10 | 15,522 | 12 | 0.30 | −1.20 | 1.04 | 2.5E-01 | 6762 |
| rs147647315 | R434H | A/G | 77 | −0.74 | 0.12 | 1.1E-10 | 13,326 | 26 | 1.04 | 0.04 | 0.52 | 9.4E-01 | 4959 |

rs150255373 (at 64366298, b37) is exclusive to EA, and rs147647315 (at 64367854, b37) has 70 of 77 copies in AA and has been linked to hypouricemia in HGMD (accession number: CM1110340). The association between rs147647315 and serum urate in AA is reported in Supplementary Data 9. The directions of association of these two variants were consistent across cohorts. rs150255373 beta: ARIC EA (-1.29), CHS (-0.91), FHS (-1.12), RS (-0.54), CoLaus (-0.97), not detected in ARIC AA, ERF, and CILENTO. rs147647315 beta: ARIC EA (-0.98), ARIC AA (-0.71), CHS (-1.60), FHS (-0.72), not detected in RS, ERF, CoLaus, and CILENTO
Alt alternate, ref reference, SE standard error, EA European ancestry, AA African-American
[a]Amino acid positions are based on the canonical form in UNIPROT (ENSP00000366797)
[b]Beta is in the unit of one SD of inverse normal transformed serum urate, which is -1.5 mg/dL

**Table 2 Exome-wide significant genes for serum urate in primary SKAT-O meta-analysis of low-frequency (MAF < 5%) putative damaging variants**

| Race | Serum urate | | | | | | Gout | | | | |
|---|---|---|---|---|---|---|---|---|---|---|---|
| | Number of variants | SKAT-O *P*-Value | Burden Beta | Burden SE | Burden *P*-Value | SKAT *P*-Value | Number of variants | Burden OR | Burden Beta | Burden SE | Burden *P*-Value |
| *SLC22A12* | | | | | | | | | | | |
| EA and AA | 97 | 1.3E-56 | −0.033 | 0.002 | 6.7E-57 | 3.98E-19 | 60 | 0.50 | −0.700 | 0.249 | 4.93E-03 |
| EA | 78 | 5.1E-42 | −0.033 | 0.002 | 2.5E-42 | 1.48E-18 | 48 | 0.41 | −0.894 | 0.322 | 5.52E-03 |
| AA | 27 | 6.4E-16 | −0.035 | 0.004 | 3.2E-16 | 8.31E-10 | 15 | 0.66 | −0.412 | 0.393 | 2.94E-01 |
| *SLC2A9* | | | | | | | | | | | |
| EA and AA | 90 | 4.5E-07 | −0.004 | 0.001 | 2.2E-03 | 2.27E-07 | 48 | 0.85 | −0.157 | 0.116 | 1.73E-01 |
| EA | 71 | 8.5E-07 | −0.006 | 0.002 | 1.5E-04 | 4.25E-07 | 36 | 0.78 | −0.249 | 0.142 | 7.92E-02 |
| AA | 28 | 3.50E-02 | −0.001 | 0.003 | 6.90E-01 | 2.00E-02 | 17 | 1.03 | 0.024 | 0.200 | 9.03E-01 |

Exome-wide significant *p*-value for SKAT-O = 1.28e-6 (= 0.05/19461 genes with 2 or more variants x2 primary SKAT-O tests)
Burden test for gout was conducted using uniform weight for estimating the effect per variant
*SE* standard error, *EA* European ancestry, *AA* African-American, *OR* odds ratio, *SKAT-O* optimal sequence kernel association test

association of rs150255373 with systolic blood pressure in the ARIC study did not yield reproducible significant associations (ARIC beta: −8.22, $p = 0.03$, replication beta: −5.04, $p = 0.16$, Supplementary Data 11).

In aggregate, the low-frequency putative damaging variants (MAF < 5%) in *SLC22A12* and *SLC2A9* were significantly associated with serum urate levels (*SLC22A12*: 97 variants, primary SKAT-O $p = 1.3 \times 10^{-56}$; *SLC2A9*: 90 variants, primary SKAT-O $p = 4.5 \times 10^{-7}$, Table 2). All 97 variants in *SLC22A12* were rare (MAF < 1%), and 82 of them were associated with lower serum urate levels (Fig. 1a). In the ARIC study, among 260 rare allele carriers, 255 carried one and 5 carried two alleles, indicating that the rare variants were largely independent. In the follow-up analysis employing separate burden test and SKAT combining all studies, the association between *SLC22A12* and serum urate levels was clearly more significant using the burden test (beta = −0.033, $p = 6.7 \times 10^{-57}$) than the SKAT ($p = 4.0 \times 10^{-19}$). The significant association of *SLC22A12* in the burden test was driven by many variants because 33 variants could sequentially be removed before the burden test result was no longer exome-wide significant. Higher CADD score and PhyloP46way primate rank score were significantly associated with larger negative effect sizes of these rare variants, supporting the potential damaging effects of these variants (CADD score beta: −0.05, $p = 2.3 \times 10^{-7}$, Fig. 1b; PhyloP46way primate rank score beta: −0.80, $p = 8.1 \times 10^{-3}$). The association between the PhastCons46way primate rank score and rare-variant effect size was also inverse, but not statistically significant (beta: −0.56, $p = 0.13$). Race-specific burden test meta-analyses showed that the aggregate effect size of the rare *SLC22A12* variants in EA and AA were of similar magnitude (EA: 78 variants, beta: −0.033, $p = 2.5 \times 10^{-42}$; AA: 27 variants, beta: −0.035, $p = 3.2 \times 10^{-16}$). Supplementary Fig. 1 presents the potentially damaging rare variants included in primary gene-based analysis in *SLC22A12* as a lollipop plot. Gout risk in carriers of these rare variants in *SLC22A12* was halved in comparison to non-carriers (burden test OR per copy of variant carried: 0.50, $p = 4.9 \times 10^{-3}$).

For *SLC2A9*, the SKAT showed stronger association than the burden test (SKAT $p = 2.3 \times 10^{-7}$, burden test: beta = −0.004, $p = 0.001$). This aggregate association was no longer exome-wide significant (SKAT $p = 3.1 \times 10^{-4}$) when the most strongly associated single-variant was removed (rs73225891; MAF 2.3%, beta = −0.15, $p = 2.8 \times 10^{-5}$ in single-variant primary meta-analysis). Secondary meta-analyses did not identify additional variants or genes associated with serum urate. All genes associated with serum urate with a SKAT-O p < $1 \times 10^{-6}$ in both

primary and secondary meta-analyses are reported in Supplementary Data 12.

**Interrogation of primary urate meta-analysis.** Among the low-frequency or rare coding variants in *SLC2A9* or *SLC22A12* that were previously reported to be associated with hypouricemia, 14 were found in our study with minor allele count ranging from 1 to 77. Of these, 12 showed associations with lower serum urate levels, and seven reached nominal statistical significance ($p < 0.05$; Supplementary Data 13). Two low-frequency coding variants in *ABCG2*, rs2231137 (V12M) and rs72552713 (Q126X), were reported to be associated with gout[18,19]. In our study, rs2231137 was not significantly associated with gout (MAF = 4%, OR = 0.91, $p = 0.46$), and rs72552713, detected almost exclusively among individuals of East Asian ancestry, was not observed in our study. Lastly, the urate- and gout-associated coding variant rs150414818 (c.1580 C > G) in *ALDH16A1*[20] was associated with urate and gout in our study (serum urate beta: 0.45 SD, $p = 0.003$, MAF = 0.16%, $n = 13,776$; gout OR = 5.93, $p = 0.02$, MAF: 0.14%, $n = 4,959$).

The interrogation of the primary SKAT-O meta-analysis results identified *HPRT1* at chromosome X as associated with serum urate levels among 34 genes linked to Mendelian form of abnormal urate levels ($p = 1.4 \times 10^{-3}$, Supplementary Data 14), the first instance that a gonosomal gene was identified as associated with serum urate in population-based studies. Other interrogations confirmed known associations for common variants in eight candidate genes. Most of these common missense variants were in high linkage disequilibrium with the known GWAS index SNPs. (Supplementary Note 1, Supplementary Data 15).

**Prioritization of rare variants in *SLC22A12* (URAT1).** To curate the large number of rare variants identified in *SLC22A12* and prioritize them for experimental studies, we used an approach that combined bioinformatic and statistic evidence to anticipate functional significance (see Methods and Supplementary Fig. 2), resulting in the identification of R325W, R405C, T467M, and K536T in the URAT1 transporter encoded by *SLC22A12* (Supplementary Data 16). Figure 2a shows the positions of these four variants in the topology of URAT1.

**SLC22A12 (URAT1) variants show reduced protein abundance.** In comparison to wild-type URAT1, all three variants R325W, R405C, and K536T showed a significant reduction in protein abundance in transiently transfected HEK293T cells (Fig. 2b,

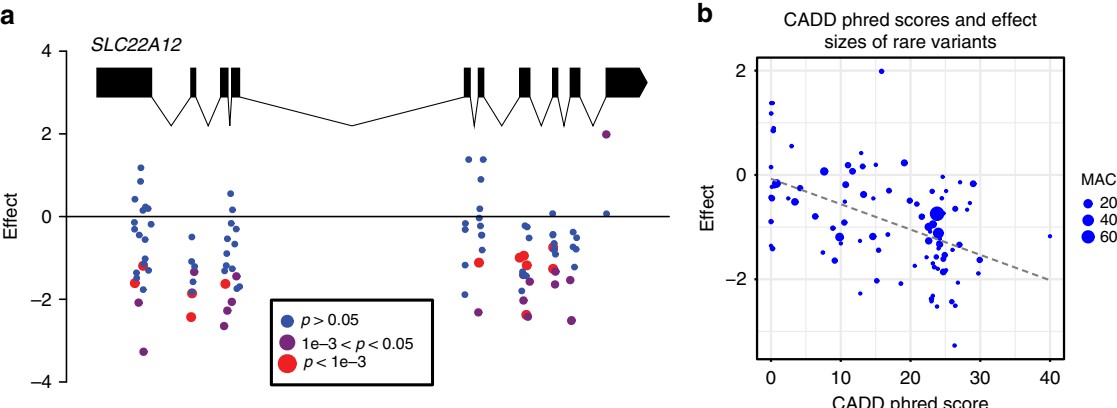

**Fig. 1** Rare putative damaging variants in *SLC22A12*. **a** Scheme of the transcript variant 1 (NM_144585.3) onto which positions of each putatively damaging variant included in the *SLC22A12* gene-based tests were mapped. Marker color reflects the association *p*-value with serum urate from the single-variant primary meta-analysis; **b** higher CADD phred score was significantly associated with larger negative effects on serum urate among the rare variants (beta = −0.05 per unit of CADD score, SE = 0.01, *p* = 2.3e-7)

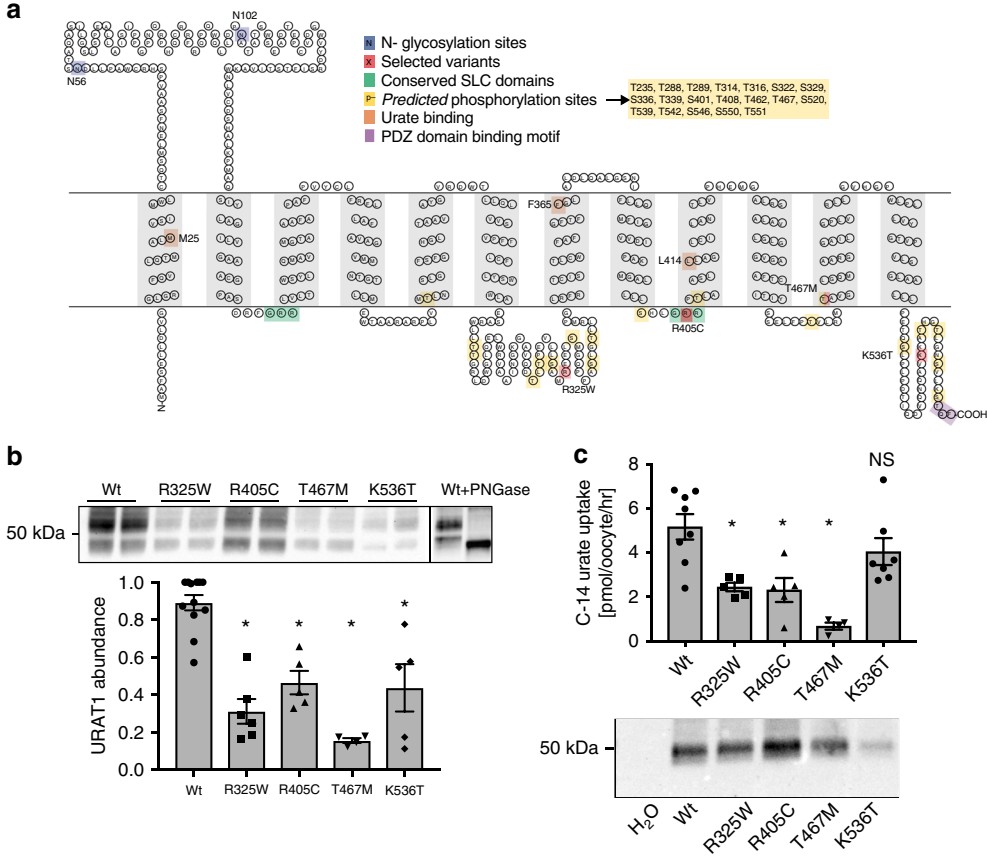

**Fig. 2** Topology and experimental results of four damaging variants in *SLC22A12*. **a** Topology of *SLC22A12* gene product URAT1 with position of variants tested marked in red. Also marked are glycosylation sites (blue); conserved domains (green); predicted phosphorylation sites (see Supplementary Methods; yellow); proposed urate-binding residues (orange); and the PDZ domain binding motif (purple). **b** Abundance of URAT1 mutant constructs in HEK293T cells, glycosylation revealed with PNGase, and summary data; *n* = 4 to 12; ±SEM; *\*p* < 0.0001 (ANOVA with Dunnett's multiple comparison test). Each lane represents a separate experiment run, processed, and quantified in parallel. The line in the image denotes the separation between two separate blots. **c** Summary data from *Xenopus* oocytes accumulation assay using C-14 labeled urate (see methods). URAT1 mediated urate transport rates with the $H_2O$ injected control transport rate subtracted (flux of the $H_2O$ injected controls was 4.3 [pmol/oocyte/hr]); *n* = 10 to 16 oocytes (processed in 5 to 8 pairs); ±SEM; *\*p* < 0.01 (ANOVA with Dunnett's multiple comparison test). Western blot shows the total abundance of each variant upon equal mRNA injection; lysates from 4–5 pooled oocytes for each construct. Note: in *Xenopus* oocytes the URAT1 monomer migrates as a single band unlike what is observed in HEK293T cells

Supplementary Fig. 3a, b). This reduced abundance was also observed for the known human disease mutant T467M and suggested that some of the molecular defect may be caused by decreased abundance of transporter protein[21]. To discern the mechanism, we first evaluated glycosylation of URAT1 as a proxy for trafficking progress. Each of the variants were maturely glycosylated, indicating a portion of the variant proteins can traffic at least as far as the trans-Golgi (Fig. 2b and Supplementary Fig. 4) and were not grossly mis-folded and completely removed via the ER-associated degradation pathway[22]. However, the ratio of maturely glycosylated to immature protein was altered for the R325W, R405C, and T467M variants as compared to the wild-type protein, suggesting a partial defect in their maturation and trafficking (Supplementary Fig. 4b). For R325W, the observation that the variant protein could gain any complex glycosylation was unexpected because of the predicted damaging nature of the rather extreme amino acid substitution in a cytosolic portion of the protein, which is infrequent across a representative mammalian alignment of URAT1 (Supplementary Fig. 5). We therefore evaluated whether transient over-expression of R325W may have obscured a mis-folded fraction, or whether R325W may instead affect the phosphorylation at the downstream S329. To test these possibilities, we mutated another arginine in the intracellular loop 3 (ICL3) that unlike R325 is not predicted to form a kinase substrate motif, R291 (Supplementary Fig. 4c, d). The R291W mutant showed very low abundance, and none of the protein trafficked to the Golgi nor acquired Endo H resistance, consistent with a severe folding variant that is recognized by the ER-associated degradation pathway quality control system[22]. The lack of an equally profound trafficking defect for R325W as compared to R291W provides initial evidence that R325W may disrupt a phosphorylation domain and regulation site of the ICL3.

### SLC22A12 (URAT1) variants display altered urate transport.

The mutant URAT1 constructs were expressed in *Xenopus* oocytes. Transport function was evaluated using a C-14 radiolabeled urate accumulation assay with a low Cl⁻ extracellular bath to maximize URAT1 dependent flux. *Xenopus* oocytes expressing R325W, R405C, or the known positive control T467M displayed significantly less C-14 urate accumulation than oocytes expressing the wild-type transporter (Fig. 2c), consistent with the variants causing a loss-of-function. Mean urate accumulation of K536T in oocytes was not significantly different from that expressing the wild-type transporter (Fig. 2c). This was unexpected because human variant carriers showed significantly higher serum urate levels. We therefore investigated transporter abundance for all variants and found clearly reduced transporter abundance of K536T (Supplementary Figs. 3c, 6a). Normalizing urate uptake for transporter abundance showed significantly higher transport rates across three time points and three different urate concentrations by K536T compared to wild-type expressing oocytes (Supplementary Fig. 6b). These observations may suggest that K536T represents a gain of transport function mutant, which is consistent with its association with higher urate levels in our association study and could reconcile the discrepancy between the observation in humans and the uptake studies in oocytes. These findings need however to be interpreted with caution, given that protein abundance could not be assessed in human target tissue. Together, these four rare variants along with the previously described R434H that was also detected in our study explained 0.8% of the urate phenotypic variance.

### Characterization of SLC2A9 (GLUT9) variants.

We employed a complementary approach for evaluating potential functional consequences of the rare aggregated and urate-associated variants in *SLC2A9* to identify functional areas in the encoded transporter GLUT9 that may specifically accumulate rare variants. Capitalizing on the known structure of a homologous transporter (see Methods), we mapped *SLC2A9* variants with absolute effect sizes > 1.5 standard deviations on a structural model of GLUT9/SLC2A9b (Fig. 3). Supplementary Fig. 7 shows all variants with absolute effect > 1 standard deviations on the structural model. Figure 4 presents the putative damaging variants with MAF < 5% based on the amino acid position of SLC2A9b. Their localization revealed three hotspots of potential functional significance: the extracellular interface (N61S; H68Y; S129L; D377N), the intracellular interface (K109N; G249S; Y478X), and the central core region (F88L; V184M; V428I; N433S). Both the extracellular and intracellular interface groups include variants in highly conserved residues with predictable damaging effects. For example, K109N at the intracellular interface is part of the canonical GRR/K motif in the short intracellular linker between the second and third transmembrane helices (Supplementary Fig. 8a, b). Variants to these or homologous residues in the second half of the protein (between TM8 and 9; e.g. R380W) cause loss of urate transport function, as do similar variants in the GRR motif in other SCL transporters (e.g., the SLC22A12 R405 variant)[23,24]. Similarly, N61S on the extracellular interface substitutes the conserved N-glycosylation site (NXS) homologous across all type I and type II GLUTs[25], and the loss of this asparagine causes significant alterations in affinity and function of related SLC2A transporters[26]. Finally, S129L represents a highly disfavored substitution, potentially impacting the position of the neighboring C128 residue previously shown to be important for determining urate affinity and transport capacity[27].

A group of particularly high interest includes the variants mapping to the predicted core helices (F88L; V184M; V428I; N433S). Previously, Long et al[28]. defined four residues, Y42, Q299, G402, and N429, as forming hydrogen bond interactions with urate in the binding pocket (with the N-O bond distance as being the typical 3.0 Å; Fig. 3b, c). The distances between the atoms in the modeled open configuration are altered by each of the 4 core variants (Fig. 3d, e; Supplementary Fig. 9) resulting in the loss of one or more of the four predicted hydrogen bonds between urate and the binding pocket of GLUT9. Interestingly, each amino acid substitution (88L; 184M; 428I; 433S) is either favored or neutral, suggesting they may be tolerated within the structure and yet change subtly the relationship of key residues in the substrate binding pocket potentially preventing the urate from binding or binding with altered affinity.

### Discussion

Using large-scale WES association studies among up to 19,517 EA and AA individuals, we confirmed common variants associated with serum urate, eGFR and UACR and identified rare urate-associated variants that mapped into loci previously identified through GWAS[11,13]. The effect sizes of rare urate-associated variants correspond to 1.0 to 1.7 mg/dL lower serum urate levels, which is more than three times larger than the effect size of common variants identified in GWAS[13]. Strikingly, dozens of rare potentially damaging variants in *SLC22A12* and *SLC2A9* were strongly associated with serum urate levels in aggregate, providing novel insights into the genetic architecture of urate metabolism. The majority of the rare variants in *SLC22A12* were associated with lower serum urate levels, and carriers showed half the risk of gout compared to non-carriers. Functional characterization of four rare variants in *SLC22A12* confirmed R325W, R405C, and T467M as loss-of-function variants. Mapping of *SLC2A9* variants onto the modeled structure of the encoded GLUT9 protein led to the identification of mutation hotspots in specific functional domains.

The preponderance of urate lowering rather than urate-increasing variants in URAT1 in our study is consistent with a

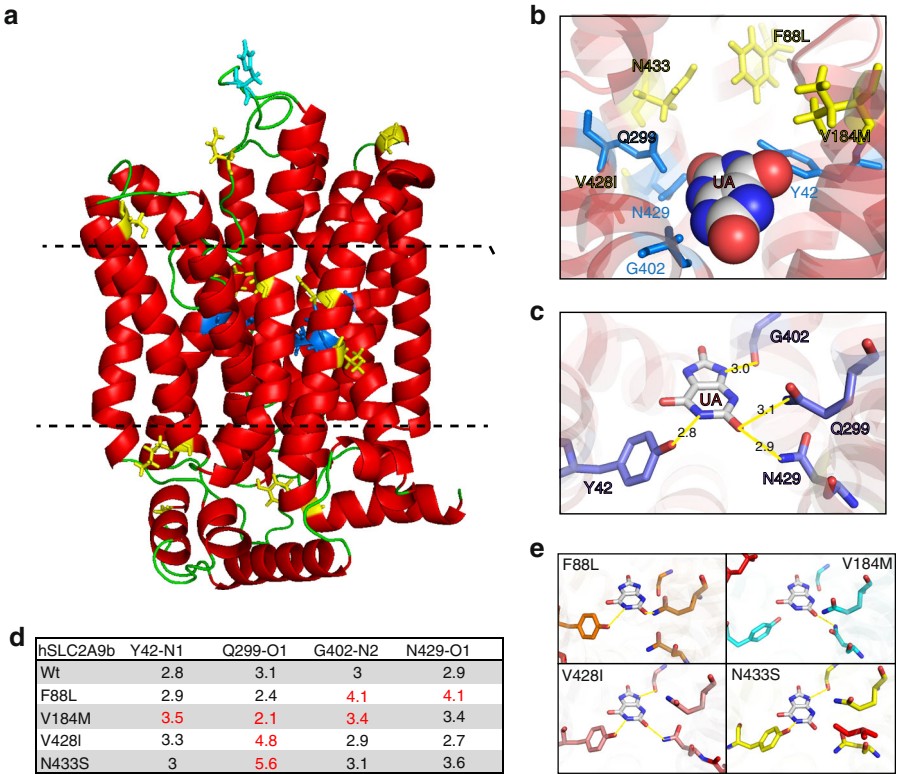

| hSLC2A9b | Y42-N1 | Q299-O1 | G402-N2 | N429-O1 |
|----------|--------|---------|---------|---------|
| Wt | 2.8 | 3.1 | 3 | 2.9 |
| F88L | 2.9 | 2.4 | 4.1 | 4.1 |
| V184M | 3.5 | 2.1 | 3.4 | 3.4 |
| V428I | 3.3 | 4.8 | 2.9 | 2.7 |
| N433S | 3 | 5.6 | 3.1 | 3.6 |

**Fig. 3** Mapping of residues to predicted model of SLC2A9b. **a** Model of human SLC2A9b in the outside open configuration, based on the rat SLC2A5 (GLUT5) crystal structure PDB:4YBQ (see Methods), all residues depicted are Wt residues. Yellow sticks = position of variants associated with lower serum urate (Wt residue shown), cyan = position of variants associated with higher serum urate (Wt residue shown), dark blue = residues predicted to be important in urate binding. Dashed lines define the functional zones, top is the extracellular interface, bottom is the intracellular interface and the middle is the transport core. **b** Visualization of the urate-binding pocket within the transporter core and the position of variants (Wt residue shown) and key urate-binding residues (Y42, Q299, G402, N429; blue). **c** Prediction of urate occupancy in the binding pocket and the length and position of predicted hydrogen bonds (yellow; in Angstroms). **d** Table of distances between critical urate interacting atoms (N, Nitrogen; O, Oxygen; see Supplementary Fig. 9) and their predicted binding partner in the presence of variant residues. Red color denotes predicted loss of hydrogen bond (PyMol polar contact prediction tool; in Angstroms). **e** Predicted hydrogen bonds (yellow) between urate and the residues of the binding pocket (mutant residues in red)

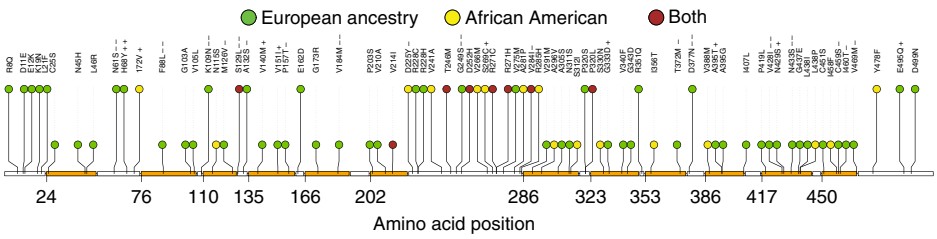

**Fig. 4** Putative damaging variants with MAF < 5% in *SLC2A9* (hSLC2Ab). Of the 90 variants included in primary gene-based test, 87 were missense, and 3 were splice variants. Of the 87 missense variants, 77 were found in the hSLC2A9b. The ± signs indicate variants selected for mapping onto the SLC2A9 predicted structure. Beta between −1.5 and −1.0 (−); beta < −1.5 (−−); beta between 1.0 and 1.5 (+); beta >1.5 (++), Orange: predicted transmembrane domains. Amino acid position based on NP_001001290 (ENSP00000422209)

previous Japanese sequencing study that identified seven exonic *SLC22A12* variants in patients with low serum urate levels. However, only one variant was validated in transport studies[29]. The large number of loss-of-function variants observed in our study might be explained by a mechanism that counter-acts the loss of uricase in humans. The loss of uricase function in humans results in higher serum urate levels compared to most other species, which was speculated to have evolutionary benefits because of several independent mutational events resulting in the loss of uricase function during hominoid evolution[30]. Another explanation for the large number of potential loss-of-function variants in *SLC22A12* and the observation that no rare variants of large effect were identified for the other studied kidney function

traits, eGFR, and UACR, could be that these variants have not been under selection pressure. This is supported by the observation that the number of rare exonic variants (MAF < 0.1%) in *SLC22A12* did not differ significantly from expectation in the ExAC database[31]. Tolerance to loss of *SLC22A12* function may be due to the presence of several other transporters that can reabsorb urate from the tubular lumen such as *SLC22A11* (OAT4) and *SLC22A13* (OAT10)[32,33]. Alternatively, gain-of-function variants in URAT1 may not be identified as readily either because they may have pathogenic effects in early development, or because individuals with gain-of-function variants may receive urate-lowering treatment and hence their serum urate levels might not reflect their genetic predisposition. We therefore expect that

genetic studies among ascertained populations with detailed information on urate-lowering therapies may yield complementary insights.

In GWAS of serum urate, *SLC22A12* is a locus of modest effect size[13]. However, in our rare-variant association study, *SLC22A12* emerged as a gene containing multiple rare variants with large effects, with five rare and experimentally confirmed variants explaining 0.8% of the serum urate variance. In contrast, *SLC2A9* is the locus with the largest effect size in GWAS of serum urate, explaining ~60% of the genetic effects attributed to all GWAS index SNPs in EA[34], but there were comparatively few rare variants of large effects identified in our sequencing association study. These findings are consistent with a genetic architecture of serum urate that is dominated by a small number of genes, namely common variants in *SLC2A9* and *ABCG2* and multiple rare variants in *SLC22A12*. The relative contributions of common and rare variants in these genes need to be updated as additional rare variants identified in our and other studies are experimentally confirmed. As a complex trait, the genetic architecture of serum urate additionally involves a large number of genetic variants with modest effects from many genes. The comparison of the genetic architecture of serum urate to those of other complex traits and diseases is currently limited by the paucity of large-scale sequencing association studies of comparable size.

Variants identified through genetic association studies in humans are often not translated into mechanistic insights on a molecular level. We tried to address this common limitation by the functional study of rare *SLC22A12* variants predicted to result in impaired URAT1 function, which provided new information on the human URAT1 molecule including new potential target residues for pharmacological modulation of urate reabsorption. We confirmed the previously described T467M variant as a loss-of-function variant[21]. Using the same methods, we identified R325W and R405C as loss-of-function and present first evidence indicating that K536T may show increased transport rates but coupled with decreased abundance. Because of the use of model systems to quantify protein abundance, further interpretation requires understanding of how the K536T variant manifests in the human target tissues.

The R405C variant is located in the second GRR motif conserved across the SLC transporter family and is proposed to be important for positioning the helix that contains the F365 residue, critical for urate-binding affinity[23,35,36]. This is supported by previous work describing the neighboring R406C as a loss-of-function human disease mutant[37]. Conversely, the R325W substitution resides in a cytosolic linker region known to harbor multiple phosphorylation sites that regulate function in closely related transporters[38]. In hURAT1, the downstream S329 residue is a predicted phosphorylation site, making R325 a potential critical residue in the substrate recognition motif (REELS) for the phosphorylating kinase. At this position, glutamine is more frequent across a representative mammalian alignment of URAT1, excepting representatives of primate linage. Given the differential affinity and transport capacity for urate between the apes and other mammals, these discrepant amino acids at position R325 may indicate a key alteration in human URAT1 function or regulation. The URAT1 K536T variant occurs in the C-terminal cytoplasmic tail in an area critical for protein-protein interactions (PDZ binding motif, S/T-X-Φ) and modifications critical for function[39]. Although the C-terminal tail is not well conserved across human MTF family members, the SLC22A12 C-terminus is highly conserved across mammalian species. Like R325 and R405, K536 is predicted to form a kinase substrate motif for targeting phosphorylation at T539 and thus could be critical for proper regulation[40]. This is supported by our findings that the K536T substitution shows significantly altered

abundance when heterologously expressed in either oocytes or HEK293T cells.

Mapping of the SLC2A9 variants on to the modeled protein structure, a complementary approach to transport studies, revealed three new potentially key regulatory and functional domains in GLUT9. We found both the extracellular and intracellular interfaces to be enriched with variants of large effect size, strongly supporting the role of glycosylation at N61 as critical for GLUT9 function. In addition we found that the four core variants, F88L, V184M, V428I, and N433S each disrupted predicted hydrogen bond interactions between urate and four key urate-binding residues (Y52, Q299, G402, and N429[28,41]), changes that may alter the substrate binding kinetics and affinity.

Clinically, the identification and experimental confirmation of functional variants in URAT1 delivers insights into important regulatory mechanisms of the protein. Insights into the regulation of URAT1 activity and abundance at the molecular level can represent a basis for the development of targeted drugs to lower serum urate. Both hyperuricemia and gout are common conditions, and gout is commonly undertreated[42]. A clinical trial of febuxostat and allopurinol, the most commonly used drug to lower serum urate, showed that only 21% of patients randomly assigned to 300 mg/d allopurinol achieved optimal serum urate concentrations[43]. The development of additional urate-lowering drugs is therefore of high clinical interest. The first URAT1 inhibitor, lesinurad, has just been approved for gout patients with treatment-resistant hyperuricemia[44]. The validated loss-of-function variants we identified in URAT1 illustrate the therapeutic potential of lesinurad and may provide insights into how to further develop this medication class. Our results may also aid in the development of SLC2A9 inhibitors to lower serum urate.

The strengths of our study include the use of WES, which enabled the detection of rare variants not limited by reference panels, poor imputation quality, or representation on a microarray. Another strength is our experimental approach that used transport assays as well as abundance and glycosylation studies in mammalian cells to provide additional insights into the molecular mechanisms. Some limitations of our study warrant mention. Several factors may have reduced our ability to detect genetic associations. First, although whole-exome capture platforms covered most of the coding regions[45], gaps in coverage may have limited our ability to detect genetic variants in those regions. Differing genotype quality filters across cohorts independent of our phenotypes could also have reduced our ability to detect true associations. Although this study represents the largest WES study of serum urate and measures of kidney function to date, the sample size was still limited for identifying rare variants that individually show significant associations with the evaluated traits. In addition, the effect size of low-frequency variants associated with kidney function may be smaller than those associated with serum urate due to imprecision in GFR estimates and higher within-person variability[46–48]. Very rare or private alleles of large effect may be more readily identified in referred patients. In addition, despite WES, a previously replicated and validated urate-lowering missense variant, rs12800450, was not present in our sample[49]. The upcoming data from the large, trans-ethnic TOPMed Program will provide an additional opportunity to further investigate the frequency of this variant in whole genome sequence data. In the present study, while we confirmed that the variants most likely to be damaging were indeed loss-of-function variants, the *SLC22A12* and *SLC2A9* variants were too numerous to allow for comprehensive experimental testing. If some of the rare variants in *SLC22A12* and *SLC2A9* were nonfunctional, however, this would lead to an underestimate of the aggregated effect because the gene-based tests combined all rare putative damaging variants based on

annotation. We focused our follow-up investigation on low-frequency and rare variants. Future functional studies are required for studying common missense variants found in known GWAS loci to evaluate their potential causal effect on serum urate levels since a missense variant in regions implicated by GWAS is not a direct support for causality[50].

In conclusion, this large-scale WES association study delivered new insights into the genetic architecture of serum urate. It discovered rare, independent urate-associated variants of large effect as well as confirmed common variants associated with serum urate, eGFR, and UACR in known GWAS loci in the general population. Dozens of rare, potentially damaging variants in *SLC22A12* and *SLC2A9* were strongly associated with serum urate levels in aggregate. Risk of gout was halved in carriers of rare variants in *SLC22A12*, illustrating the therapeutic potential of the new URAT1-blocker lesinurad. The R325W and R405C variants in URAT1 were experimentally validated as new loss-of-function variants. Mapping of *SLC2A9* variants onto the modeled GLUT9 protein structure revealed new potentially key regulatory and functional domains. These represent a basis for the design of additional urate-lowering therapies, for which there is a high clinical need given the rising burden of gout.

## Methods

**Study populations.** The study populations included 15,431 EA and 3698 AA participants from eight population-based studies: the Atherosclerosis Risk in Communities study (ARIC), Cardiovascular Health Study (CHS), Cilento study, CoLaus study, Erasmus Rucphen Family study (ERF), the National Heart, Lung, and Blood Institute GO Exome Sequencing Project (ESP), Framingham Heart Study (FHS), and the Rotterdam Study (RS). All studies were approved by their respective Institutional Review Boards, and all participants provided written informed consent. Details about the studies have been reported previously, and the key references as well as the summary of the design of each study are reported in Supplementary Data 17.

**WES and annotation.** Details on sequencing, variant calling, and variant quality control are reported in Supplementary Data 18. Briefly, WES in ARIC, CHS, and FHS was performed by the Baylor College of Medicine Human Genome Sequencing Center as a part of the Cohorts for Heart and Aging Research in Genomic Epidemiology (CHARGE) Consortium based on established protocols (Baylor College of Medicine Human Genome Sequencing Center (BCM-HGSC) protocols for sequencing library construction, https://www.hgsc.bcm.edu/content/protocols-sequencing-library-construction, and Mercury analysis pipeline, https://www.hgsc.bcm.edu/content/mercury)). The CILENTO, CoLAUS, ERF, ESP, and Rotterdam studies performed WES following similar protocols[51,52]. The annotation process combined all variants from each study and annotated variants using ANNOVAR[53] and dbNSFP v2.0[54] based on the reference genome GRCh37 and National Center for Biotechnology Information RefSeq.

**Outcomes.** The primary outcomes of this study were the kidney function related traits: serum urate, eGFR, and UACR. The methods for calculating eGFR were the same as in previous GWAS of kidney function[55]. Briefly, the MDRD 4-variable equation[47] was applied to calculate eGFR after serum creatinine levels in each study were calibrated using age- and sex-specific means from the nationally representative NHANES Survey. Prevalent gout was analyzed as a secondary outcome for significant variants or genes associated with serum urate levels. Gout cases were ascertained based on self-report. The laboratory methods for measuring serum urate, serum or urinary creatinine, and urinary albumin are reported in Supplementary Data 17.

**Association analysis.** Serum urate, eGFR, and UACR were analyzed as continuous outcomes. Due to their skewed distribution and consistent with previous studies[11,12], natural log transformation was applied to eGFR and UACR. Then residuals of serum urate, log-transformed eGFR and UACR were generated adjusting for age, sex, genetic principal components, and study center if applicable. To minimize potential influence from outliers, inverse normal transformation was applied to the residuals, which were used as the outcome in genetic association analyses. Gout was analyzed as a binary outcome adjusting for age, sex, genetic principal components, and study center if applicable. Each study generated cohort-level association summary results based on a linear regression model for serum urate, log-transformed eGFR and UACR, and a logistic regression model for gout using the seqMeta R package version 1.3. Family studies additionally accounted for relatedness. Subsequently, the association summary results from each study were combined via meta-analyses.

An overview of the workflow of all meta-analyses and secondary analyses is depicted in Supplementary Fig. 10. The primary meta-analyses included single-variant tests and the optimal sequence kernel association test (SKAT-O)[56] combining results from all EA and AA studies. The primary single-variant meta-analysis results were filtered by total minor allele count (MAC) ≥ 10. The primary SKAT-O meta-analyses included two sets of tests that aggregate low-frequency and rare putative damaging variants with overall MAF < 1% and < 5%. Putative damaging variants (predicted to change protein function as described in MacArthur et al.[57]) included those annotated as missense, coding insertion-deletion (indels), splice sites based on ANNOVAR, predicted splice sites based on AdaBoost (score ≥ 0.6), SPIDEX (absolute change of psi score ≥ 5), or a method combining adaptive boosting and random forest (score ≥ 0.6)[58–60]. Only genes with two or more putative damaging variants across all studies were included. The weight for each variant was the default beta distribution with parameters 1 and 25. Details for the secondary meta-analyses of all traits and follow-up analyses on urate-associated rare variants and blood pressure are reported in the Supplementary Methods section and Supplementary Data 19 to 23.

**Exome-wide significance threshold.** The exome-wide significance thresholds were determined by the Bonferroni method based on the number of variants or genes included in the primary meta-analysis. The significance thresholds of the primary single-variant meta-analyses were $7.8 \times 10^{-8}$ ($= 0.05/640,895$ variants) for serum urate, and $6.4 \times 10^{-8}$ ($= 0.05/781,062$ variants) for eGFR and UACR. The significance thresholds of the primary SKAT-O meta-analyses were corrected for the number of genes with at least two variants and the two sets of tests (MAF < 1% and < 5%): $1.28 \times 10^{-6}$ ($= 0.05/[19461$ genes*2]) for serum urate, and $1.28 \times 10^{-6}$ ($= 0.05/[19549$ genes*2]) for eGFR and UACR. We performed post-hoc power calculations for 80% power to detect the minimum effect size for single variants with MAF of 0.01, 1, and 5% for the three traits in the overall study population and in EA and AA separately assuming an alpha of $7.8 \times 10^{-8}$ to account for multiple testing.

**Follow-up on exome-wide significant variants and genes.** First, exome-wide significant rare variants that mapped into a previously known GWAS locus and common variants in the *SLC2A9* region were evaluated for their independent association with the outcome by conditioning on the GWAS index SNP. This analysis was conducted in the ARIC study, the largest among the contributing studies. GWAS variants were imputed based on the 1000 Genomes reference panel phase I version 3[61]. Second, exome-wide significant genes from the primary SKAT-O meta-analysis were evaluated using separate burden test and SKAT to assess the aggregate direction of association of the rare variants. Third, to determine whether the significant associations using SKAT-O were mainly driven by a small number of variants per gene, we conducted sensitivity gene-based analysis by successively removing the variants with the strongest association based on single-variant test results until the p-value of the gene-based test (burden or SKAT, whichever generated the association with the lowest *p*-value) was no longer exome-wide significant. Fourth, to gain insight into the predicted consequence of the rare putative damaging variants in exome-wide significant genes, we regressed the PhyloP46way primate rank score, PhastCons46way primate rank score, and CADD phred score of these variants against their effect sizes from the primary single-variant meta-analysis[62–64]. Finally, we calculated the urate phenotypic variance explained by the five rare urate-associated and experimentally confirmed variants from our study or a previous study (R325W, R405C, T467M*, R434H, K536T) based on their effect sizes in the primary single-variant analysis[17].

**Interrogation of known urate-associated rare variants.** We systematically searched the literature (see Supplementary Methods) for low-frequency coding variants that were reported to be associated with gout or serum urate and retrieved their association statistics from our primary meta-analysis.

Selection of variants for experimental studies. We prioritized *SLC22A12* variants that were most likely to have functional or regulatory effects based on the long isoform and the corresponding mRNA amino acid sequence (NP_653186; see protocol depicted in Supplementary Fig. 2 and Supplementary Methods). We further required that the candidate variants showed nominal association with serum urate ($p < 0.05$) and had consistent direction of association among the cohorts in the overall single-variant meta-analysis of serum urate. Among four candidate variants selected for study (Supplementary Data 16), rs200104135 (T467M) was selected as a positive control because it was reported in the Human Gene Mutation Database (HGMD) as a variant for hypouricemia (accession number: CM139310)[21].

**Molecular biology.** The human SLC22A12 (URAT1) construct in the pcDNA3.1 vector was obtained as described previously[49]. The SLC22A12 variants (as denoted in the cDNA of SLC22A12 isoform a, NP_653186.2) R325W, R405C, T467M, and K536T were created using the QuikChange site-directed mutagenesis kit (Agilent Technologies, USA) and the primers were ordered from IDT (Integrated DNA Technologies, USA, sequence in Supplementary Methods). mRNA for injection and expression in *Xenopus* oocytes was prepared from the SLC22A12 mutant

expression vectors using the mMessage mMachine Kit and the MEGAclear kit (Ambion/Life Technologies, USA) according to the manufacturer's protocol.

**Mammalian cell culture and transfection protocol.** Human embryonic kidney cells (HEK293T, GenHunter Corp, USA) were cultured in DMEM (Life Technologies/Gibco 11995) supplemented with 10% FBS and 500 mg/ml penicillin & 50 mg/ml streptomycin. Cells were grown to > 90% confluence and plasmids transfected using X-tremeGENE 9 (Roche, GER) according to manufacturer's protocol.

**Western blot protocol.** Cells were lysed 48 hours post-transfection using lysis buffer (1% deoxycholic acid, 1% triton X-100, 0.1% SDS, 150 mM NaCl, 1 mM EDTA, 10 mM Tris HCl pH 7.5, and protease inhibitor (complete protease inhibitor, Sigma, USA). The samples were incubated at 37 °C for 30 min with 5× laemmli buffer, with 5% 2-beta mecaptoethanol. The samples were run on precast 10% Stain-Free Gels (Biorad, USA), transferred to nitrocellulose membrane using the Bio-Rad Transblot transfer system (Biorad, USA), and blocked in 5% non-fat milk. The appropriate primary antibody (anti-SLC22A12; MBL International Corp (JPN), #BMP064, 1:1000) was added and the membrane was incubated overnight at 4 °C. After washing and the addition of appropriate secondary antibody the membrane was exposed with Super Signal ECL (Pierce) and the chemiluminesence signal was captured using a ChemiDoc Imagining system (BioRad, USA) and band density was calculated using BioRad's Image Lab Software. Normalization/Loading controls were done by calculating the total protein loaded in each lane using the BioRad Stain-Free Gel System. Statistical comparisons of density measurements from western blots were done with the Student's t-test for pair wise comparisons, or an ANOVA, used with a Tukey's or Dunnett's Test for multiple comparisons (Prism 7, GraphPad, USA). All reported means are ± standard error of the mean (SEM).

**Deglycosylation protocol: PNGase F.** After 48 h of transfection, the cells were lysed and total protein harvested and estimated. A volume of 10 μg of total protein was denatured in 10XG buffer by heating at 100 °C for 10 min. The denatured mix was then treated with 10XG2, 10% NP-40, PNGase F (NEB, USA), and $H_2O$ and incubated at 37 °C for 1 hr. At end of the reaction, the sample was analyzed by western blot. EndoH: 10 μL of total protein sample was denatured as above. The denatured mix was then treated with 10XG5, EndoH (NEB, USA), and $H_2O$ and incubated at 37 °C for 1 hr. At end of the reaction, the sample was analyzed by western blot.

**Xenopus oocyte cell expression.** Stage V-VI de-folliculated *Xenopus laevis* oocytes were obtained from Ecocyte Bioscience US LLC (USA). On the day of delivery, the oocytes were selected and injected (Nanoinject II, Drummond Scientific, USA) with 50 nl of either mRNA or with $H_2O$ (control). All experiments were done on the third day after injection. Oocytes after injection were cultured in a modified L-15 media (OR-3) and kept at 15–20 °C. Oocyte batches from three different shipments were used in the western blot and transport experiments.

**Western blot protocol for Xenopus oocytes.** Three days after injection, 4–5 oocytes per treatment were pooled and placed into a 1.5 ml tube. Ice-cold oocyte lysis buffer was added (in mM: 20 Tris-HCl, 140 NaCl, 2% Triton-x-100, and protease inhibitors) and the oocytes were incubated on ice for 30 min. Then the oocytes were homogenized and spun at 4 °C at 7500 RPM to remove the nucleus, yolk, and cellular debris. A second incubation period on ice was followed by a final 16,000 RPM spin to remove remaining insoluble materials. The cell lysate was then stored at −80 °C until used for the western blot. SLC22A12 antibody was used to probe the blot (anti-SLC22A12; MBL International Corp (JPN), #BMP064, 1:1000) and normalization/loading controls were done by calculating the total protein loaded in each lane using the BioRad Stain-Free Gel System. The blot was visualized and analyzed as described above.

**C-14 uric acid (urate).** Radiolabeled C-14 uric acid (Moravek INC, USA) was dissolved in a stock solution of 2 mM NaOH with a final uric acid concentration of 2 mM (either 2 mM hot-radiolabeled uric acid or 1 mM hot and 1 mM cold uric acid). For oocyte transport experiments, the radiolabeled C-14 uric acid (urate) was included at various concentrations in ND96 Ringer's solution (in mM: 96 NaCl, 2 KCl, 2 MgCl2, 5 HEPES, 1.8 CaCl2, pH 7.5) or low Cl− ND96 (in mM: 96 Na Gluconate, 2 KCl, 2 MgCl2, 5 HEPES, 1.8 CaCl2, pH 7.5).

**Urate transport assays.** *Xenopus* oocyte transport experiments were conducted as previously described[14]. Oocytes were incubated for 60 or 120 min at room temperature in a 150 μM, 300 μM, or 600 μM C-14 urate/ND96 solution. Transport rates were based on the estimated volume of a single oocyte being 1 μl. After incubation the oocytes were washed in ice-cold ND96 solution three to five times to remove any residual C-14 urate from the oocytes and pooled for scintillation counting: two oocytes were placed in each scintillation tube with lysis buffer of 200 μl 10%SDS. Three ml of scintillation fluid (SafetySolve, RPI Corp, USA) was then added and their counts per minute counted using a Beckman Scintillation counter (LS 5000TA, Beckman Coulter Inc., USA). For each batch of counting, a set of control vials of

known volume and concentration of C-14 urate was included, allowing for a calculation of C-14 urate concentrations based on the counts measured in any of the experimental samples. In a Supplementary analysis of transport rates of the SLC22A12 mutants, the rate of uptake was further normalized to the expression of SLC22A12 using western blot. Significance was evaluated using a Student's t-test comparing the mutant to the wild-type versions of the transporter.

**Protein structural modeling and bioinformatics.** SLC22A12/URAT1 topology and transmembrane domains predicted using a constrained Phobius prediction method and verified using a HMMTOP analysis[65]. The 2D topology was drawn using TOPO2 (http://www.sacs.ucsf.edu/cgi-bin/open-topo2.py). SLC2A transporter NCBI Reference Sequences were obtained from the NCBI and aligned using MacVector (MacVector INC, USA). The structural model of human SLC2A9b was constructed in a manner similar to the previously published Long et al.[28]. The Wt and variant protein sequence were submitted separately to the I-Tasser Server at the University of Michigan using the recent crystal structure of rat SLC2A5 (PDB:4YBQ) as the template[66,67]. The resulting model had a confidence score (C-score) of +0.42 (scale −5 to +2), indicating a model with a relatively high confidence based on template alignment and structural similarity. The modeling of urate as a substrate and the SLC2A9b binding pocket with predicted hydrogen bonding interactions was based on published docking studies by Long et al., and putative damaging variants with urate-associated betas < −1 and > 1 were mapped onto the hSLC2A9 model (Supplementary Data 24)[28].

**Code availability.** The analysis plans for cohort-level association analyses, meta-analyses, candidate gene interrogation, and replication of rs150255373 are included in Supplementary Note 2. The programs that implemented these analysis plans are available upon request.

## Data availability
The summary statistics of the meta-analyses have been submitted to dbGaP accession phs000930 and can be requested from there.

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

## Acknowledgements

The authors gratefully acknowledge the resources generated by the NHLBI GO Exome Sequencing Project. A full list of Project members and funding information is found in Supplementary Note 3. Grants supporting individual investigators are listed below. Information about funding of the contributing parent studies is included in Supplementary Note 3. The work of Y.L. and A.K. was supported by KO 3598/4-1, and the work of A.K. additionally by KO 3598/3-1, CRC 1140, and CRC 992 (all German Research Foundation). The work of O.M.W. was supported by NIDDK R01DK114091 and American Heart SDG grant 14SDG18060004. Z.K. received financial support from the Leenaards Foundation, the Swiss Institute of Bioinformatics, and the Swiss National Science Foundation (31003A-169929) and SystemsX.ch (51RTP0_151019). A.D. is supported by a Veni grant (2015) from ZonMw. A.D., J.L., and C.M.vD. have used exchange grants from Personalized pREvention of Chronic Diseases consortium (PRECeDI) (H2020-MSCA-RISE-2014). The views expressed in this manuscript are those of the authors and do not necessarily represent the views of the National Heart, Lung, and Blood Institute; the National Institutes of Health; or the U.S. Department of Health and Human Services.

## Author contributions

A.Tin, A.G.U., M.C., P.V., C.M.vD., E.B., B.M.P., C.S.F., O.M.W., and A.K. contributed to study design. B.M.P., Z.K., P.V., A.D., A.G.U., M.C., and C.M.vD. contributed to study management. P.V., M.C., A.G.U., R.S., and M.C. contributed to subject recruitment. A. Tin, O.M.W., C.R-C., J.E.H., S.S., A.M., Z.K., A.Y.C., A.Teumer, S.G., Y.L., A.D., A.M., J. A.B., B.M., N.A., A.D., R.S., T.N., Q.Y., M.-H.C., M.S., B.Y., J.L. and A.K. contributed to

statistical methods and analysis. E.B., J.vR., R.K., and A.G.U. contributed to sequencing. A.Tin, O.M.W., Z.K., A.M., J.vR., R.S., Y.L., R.K., A.M., J.A.B., N.A., A.M., T.N., and J.L. contributed to Bioinformatics. M.L. and O.M.W. contributed to experimental work. A. Tin, O.M.W., C.R.-C., J.E.H., S.S., A.Y.C., Y.L., A.D., A.M., J.A.B., B.M., A.D., R.S., C.M. vD, and A.K. contributed to the interpretation of the results. A.Tin, Y.L., O.M.W., and A. K. contributed to the drafting of the manuscript: A.Tin, O.M.W., C.R.-C., J.E.H., S.S., B. M.P., Z.K., A.M., A.Y.C., A.Teumer, S.G., M.C., B.Y., Y.L., A.M., J.A.B., B.M., A.G.U., A. D., T.N., M.C., C.M.vD., O.M.W., and A.K. contributed to the critical review of manuscript.

### Additional information

**Competing interests:** B.M.P. serves on the DSMB of a clinical trial funded by the manufacturer (Zoll LifeCor) and on the Steering Committee of the Yale Open Data Access Project funded by Johnson & Johnson. A.Y.C. and C.S.F. are currently employed by Merck Research Laboratories. The remaining authors declare no competing interests.

Adrienne Tin[1], Yong Li[2], Jennifer A. Brody[3], Teresa Nutile[4], Audrey Y. Chu[5,6], Jennifer E. Huffman[5,7], Qiong Yang[6,8], Ming-Huei Chen[5,6], Cassianne Robinson-Cohen[9], Aurélien Macé[10], Jun Liu[11], Ayşe Demirkan[11,12], Rossella Sorice[4,13], Sanaz Sedaghat[11], Melody Swen[1], Bing Yu[14], Sahar Ghasemi[15,16], Alexandra Teumer[15,16], Peter Vollenweider[17], Marina Ciullo[4,13], Meng Li[18], André G. Uitterlinden[19], Robert Kraaij[19], Najaf Amin[11], Jeroen van Rooij[19], Zoltán Kutalik[20], Abbas Dehghan[11], Barbara McKnight[3], Cornelia M. van Duijn[11,21], Alanna Morrison[14], Bruce M. Psaty[3], Eric Boerwinkle[14], Caroline S. Fox[5,6], Owen M. Woodward[18] & Anna Köttgen[1,2]

[1]Department of Epidemiology, Johns Hopkins Bloomberg School of Public Health, Baltimore, MD 21205, USA. [2]Institute of Genetic Epidemiology, Faculty of Medicine and Medical Center, University of Freiburg, Freiburg 79106, Germany. [3]Cardiovascular Health Research Unit, DoM, University of Washington, Seattle, WA 98195, USA. [4]Institute of Genetics and Biophysics "Adriano Buzzati-Traverso" - CNR, Naples 80131, Italy. [5]Population Sciences Branch, Division of Intramural Research, National Heart, Lung, and Blood Institute, Framingham, MA 01702, USA. [6]Framingham Heart Study, National Heart, Lung, and Blood Institute, Framingham, MA 01702, USA. [7]VA Boston Healthcare System, Center for Population Genomics, Jamaica Plain, MA 02130, USA. [8]Department of Biostatistics, School of Public Health, Boston University, Boston, MA 02118, USA. [9]Department of Nephrology, Vanderbilt University Medical Center, Nashville, TN 37232, USA. [10]Department of Computational Biology, University of Lausanne, Lausanne 1015, Switzerland. [11]Department of Epidemiology, Erasmus Medical Center, Rotterdam 3000 CA, Netherlands. [12]Department of Human Genetics, Leiden University Medical Center, Leiden 2300 RA, The Netherlands. [13]IRCCS Neuromed, Pozzilli 86077 Isernia, Italy. [14]UTHealth School of Public Health, Houston, Texas 77030, USA. [15]Institute for Community Medicine, University Medicine Greifswald, Greifswald 17475, Germany. [16]Partner site Greifswald, German Center for Cardiovascular Research (DZHK), Greifswald 17475, Germany. [17]Department of Internal Medicine, Centre Hospitalier Universitaire Vaudois (CHUV), Lausanne 1011, Switzerland. [18]Department of Physiology, University of Maryland School of Medicine, Baltimore, MD 21201, USA. [19]Department of Internal Medicine, Erasmus Medical Center, Rotterdam 3000 CA, The Netherlands. [20]Institute of Social and Preventive Medicine (IUMSP), Centre Hospitalier Universitaire Vaudois (CHUV), Lausanne 1010, Switzerland. [21]Leiden Academic Centre for Drug Research, Leiden University, Leiden 2300 RA, Netherlands. These authors jointly supervised this work: Owen M. Woodward, Anna Köttgen.

