## [Peer Review File · Nature Communications]

Reviewers' Comments:

Reviewer #1:

Remarks to the Author:

This is a large exome-based genome-wide study in renal phenotypes (urate, eGFR, UACR) with follow-up functional analysis of variants detected in the SLC22A12 gene and modelling to predict functional impact of SLC2A9 variants. The title led me to expect more from the eGFR/UACR side of things, however the focus was very much in urate, driven by the findings of SLC2A9 and SLC22A12. I have no major criticism of the way the genetic aspect of the study was carried out, it was a well rationalised - a genome-wide assessment of the impact of lower frequency more penetrant missense / nonsense / splicing variants on urate etc is a well rationalised study and the findings, for urate at least, are interesting. My major criticism is the structure of the paper. It is very complex, with a great amount of data presented (e.g. 25 Supp Tables) and could benefit from being re-organised, perhaps focusing entirely on urate as a phenotype. This would allow some other interesting findings in the urate genes to be explored in more detail. They are established loci, so further exploration here needn't be constrained by achieving exome-wide significance (Certainly the findings, to me, emphasize the relative lack of genetic heterogeneity in urate control compared to renal function.)

Specific comments:

1. Carrying on from my general comment above:

- a. the findings of ABCG2 are interesting in the context of recent burden papers (Stiburkova Rheumatology and Higashino RMD Open). Can the authors explore this further (I acknowledge it would be unreasonable to expect functional studies). What is the non-rs2231142 burden odds ratio?
- b. Can SLC17A1/A4 be explored more, at least from a bioinformatic, structural modelling perspective. Also GCKR.
- c. INHBC is interesting. These data pretty much confirm it as causal at the Kottgen 2013 GWAS locus. Again, can this be explored further? Especially in the context that essentially nothing is known about the function of this gene in urate control.
- d. WDR1 and ZNF518B - are these independent from SLC2A9? Conditional analysis should be done.

2. While the association data for rs150255373 with blood pressure are interesting, they are buried in the paper and not specifically mentioned in the main text. Either remove or report in main text.

3. Line 291, incorrect Supp Figure referred to.

4. Line 338-339. That the four SLC22A12 variants explain 0.8% of phenotypic variance is remarkable. Could the authors please calculate the phenotypic variance explained by all coding variants listed in Table 15B.

5. Throughout be consistent between the use of uric acid / urate. The latter is preferable.

6. Generally, it is preferable to use the word 'genetic variant / variation' rather than mutation.

7. Lines 389-390. Here it is implied that individuals taking urate-lowering therapy were included in the analysis. This would be far from ideal. Please confirm that this wasn't the case. If it was, then such individuals should be excluded.

8. That the rs12800450 variant originally reported by Tin et al was not detected in this study is extraordinary. HapMap predicts an allele frequency of ~1% in people of European and African ancestry. Therefore for this variant to be not present at all does imply an error (technical or analytical) at some point, that may be specific for this variant, or more general.. Could the authors

please investigate this further.

9. Figure 1. Panel B appears self-serving. The rare variants were originally selected as $\beta < 0$ and $\beta > 0$ for inclusion in the plot. Panel C - why not plot the absolute value of β ?

Reviewer #2:

Remarks to the Author:

The authors conducted an exome-wide association study of kidney function-related traits, using ~20,000 European and African subjects. By applying a set of association analyses targeting not only common variants but also low-frequency or rare alleles. Gene-based test identified rare variant risk on uric acid levels at SLC2A9 and SLC22A12 genes. Subsequent analysis validated functional roles of these variants on the traits. This is a well-designed and powered genetic study to assess roles of rare variants on human complex traits, and this reviewer has few comments.

While the authors assessed several kidney function-related traits, significant signals were only observed at uric acid. This may imply relatively larger impacts of rare variants on uric acid rather than other traits. Discussions on why it happens may be useful (e.g. selection pressure, functional character of transporter genes, etc).

Variant associations of SLC2A9 and SLC22A12 on uric acid has been reported in many populations including east Asians, which should be mentioned.

Reviewer #3:

Remarks to the Author:

Well written manuscript with information clearly presented, particularly in the supplementary material. This manuscript describes exome resequencing studies in up to 15,431 individuals of European and 3,698 individuals of African American descent, which identified multiple low frequency SNPs associated with serum urate and having functional effects. Clarification of several questions would improve evaluation of the data described in this manuscript.

1. The demographics and biochemistry measurements reported for participants in S10A raises questions as to whether the MDRD-4v equation is best suited to estimate GFR in these individuals, particularly older participants; for 5+ years the CKD-EPI equation has been commonly used by many large commercial clinical labs for eGFR reporting. CKD-EPI reduces bias where $eGFR > 60$ and was recommended by KDIGO 2013. The mean $eGFR_{cr}$ in each cohort included in this study is > 70 ; does using different equations to calculate eGFR affect relevant results? Was serum urate levels measured outside an acute flare for relevant participants and were participants taking any urate lowering medication?

2. Please include a matrix summarizing the power of this study to identify race specific effects for low frequency variants for the primary phenotypes in the methods. For the aggregated results, was the effect of having specific combinations of SNPs considered? Were any of the proposed-risk rare variants in SLC22A12 and SLC2A9 identified in control populations?

3. Multiple capture methods have been used to enrich the exons – what is covered and how many exons / entire genes are missed based on each approach? Were many miRNA genes covered by the selected capture approaches?

4. The QC filters for original data calling are different between the groups; was this apparent during the analysis e.g. were SNP calls (MAF / MAC rates) consistent between cohorts? What minimum read depth was used for each cohort when calling variants? What was the average

number and range of SNPs identified per exome in each cohort? What does 'swaps fixed' mean?

5. SNPs have been previously reported as associated with [serum urate] based on imputed and directly typed GWAS data; e.g. from >140,000 individuals in which many cohorts used minor allele frequency <1% as part of their QC filter. For cohorts that have both exome sequencing and high density imputed data, would the SNPs ($MAF > 1 \times 10^{-5}$) have been identified regardless of which approach was used for primary genotyping? i.e. does the high density imputation data for these lower frequency variants equate to the NGS results? From the 16 variants identified with exome-wide significance (line 268), 14 were common – were these all previously identified by larger GWAS for serum urate levels? How many novel SNPs ($MAF > 1 \times 10^{-5}$; $MAF \leq 1$) were identified by WES that were not available in the 1 KGPv3 reference imputation data.

6. For the listed SNPs with most evidence of significance based on a single variant test or burden analysis, was significance identified and a consistent direction of effect observed across all cohorts?

7. It is surprising that a whole genome sequencing study identifying lower frequency variants associated with gout and serum uric acid levels was not referenced in this manuscript; was there any trend towards association in this study for Sulem and colleagues' low-frequency missense variant (c.1580C>G in ALDH16A1) associated with gout (OR = 3.12, P = 1.5×10^{-16} , at-risk allele frequency = 0.02) and serum uric acid levels (effect = 0.36 P = 4.5×10^{-21} (Nat Genet. 2011 43(11):1127-30. doi: 10.1038/ng.972).

8. Given the traditional increased risk of being male for these phenotypes, was sex-specific meta-analyses considered for all genes? Were reference ranges for lab tests different for males and females? If so, was this considered in the relevant association analyses?

Minor comments:

a) I do not see any data or code availability statements in the manuscript.

b) It would be helpful in supplemental Figure S1 to include numbers of individuals / snps / genes considered at relevant stages.

c) Several of the references are incomplete e.g. ref 35 Voorman et al.,

d) There are font changes within the supplementary tables e.g. S1B cell E4 vs E5 is Arial -> Calibri

e) Is it possible to include the direction of effect in each cohort for SNPS reported from the meta-analyses in supplementary material?

Response to Reviewers' comments:

We would like to thank the Reviewers for the constructive review of our manuscript. We have carefully addressed each of the comments below and in the revised manuscript. In the resubmitted version, the changes in response to the comments are highlighted in red.

Reviewer #1 (Remarks to the Author):

This is a large exome-based genome-wide study in renal phenotypes (urate, eGFR, UACR) with follow-up functional analysis of variants detected in the SLC22A12 gene and modelling to predict functional impact of SLC2A9 variants. The title led me to expect more from the eGFR/UACR side of things, however the focus was very much in urate, driven by the findings of SLC2A9 and SLC22A12. I have no major criticism of the way the genetic aspect of the study was carried out, it was a well rationalised - a genome-wide assessment of the impact of lower frequency more penetrant missense / nonsense / splicing variants on urate etc is a well rationalised study and the findings, for urate at least, are interesting.

Response: We thank the Reviewer for this positive feedback.

My major criticism is the structure of the paper. It is very complex, with a great amount of data presented (e.g. 25 Supp Tables) and could benefit from being re-organised, perhaps focusing entirely on urate as a phenotype. This would allow some other interesting findings in the urate genes to be explored in more detail. They are established loci, so further exploration here needn't be constrained by achieving exome-wide significance (Certainly the findings, to me, emphasize the relative lack of genetic heterogeneity in urate control compared to renal function.)

Response: We understand the Reviewer's point and have changed the title to "Large-Scale Whole-Exome Sequencing Association Studies Identify Rare Functional Variants Influencing Serum Urate Levels". In addition, we have shortened the text related to the association studies of eGFR and UACR, and removed two supplemental tables.

Nine of the 24 supplemental tables currently in the paper (one was added in response to a comment from **Reviewer #3**) are devoted to documenting method details for enhancing the reproducibility of our study. Although many of these could be converted to free text, we would like to retain the table format because free text may not be as helpful to the reader as the table format to locate this information.

Given that our results represent the largest exome sequencing association study of eGFR and UACR to date, we believe that these results are worth reporting although no associations in novel loci were identified. We have therefore kept this information in the supplemental tables and text.

Specific comments:

1. Carrying on from my general comment above:

a. the findings of ABCG2 are interesting in the context of recent burden papers (Stiburkova Rheumatology and Higashino RMD Open). Can the authors explore this further (I acknowledge it would be unreasonable to expect functional studies). What is the non-rs2231142 burden odds ratio?

Response: Given that the two references above used gout as the outcome and the Reviewer asked for an odds ratio, we assume that the Reviewer referred to gout as the outcome. Without rs2231142, the putative functional variants in ABCG2 did not show a significant association with gout in our study (burden OR=1.00, p=0.80). We also specifically investigated the variants implicated in Stiburkova *et al.* 2017 (p.V12M, rs2231137, OR=0.91, p=0.46, MAF=0.04) and Higashino *et al.* 2017 (Q126X, rs72552713, not detected in our population consistent with its

absence among European ancestry individuals in the large ExAC database). We have added these results to the main text (**Methods, page 8, paragraph 2; Results, page 15, paragraph 3**).

b. Can SLC17A1/A4 be explored more, at least from a bioinformatic, structural modelling perspective. Also GCKR.

Response: As requested by the Reviewer, we have now performed structural modeling of *GCKR* to investigate rs1260326, encoding the amino acid substitution P446L. We used the crystal structure for human *GCKR* (PDB 4BB9¹) to create a structural model of *GCKR* using PyMOL. We mapped the P446 residue (red) and the key domains for binding Fructose -6- phosphate (blue) and interactions with GCK (yellow). P446L does not map into the previously identified key interaction domains.¹ (**Reviewer Figure R1**)

For the SLC17A1 or SLC17A4 protein, no templates have been published that would allow us to create a homology model of the structure and to present predictions of how genetic variants may affect the function of these transporters.

Figure R1. Structural model of GCKR. P446 residue (red), the key domains for binding Fructose -6- phosphate (blue) and interactions with GCK (yellow).

c. *INHBC* is interesting. These data pretty much confirm it as causal at the Kottgen 2013 GWAS locus. Again, can this be explored further? Especially in the context that essentially nothing is known about the function of this gene in urate control.

Response: We agree with the Reviewer that it is tempting to speculate about a role of *INHBC* as the causal gene at the locus. However, in this locus, the association signal in the region is driven by a common variant in our study: the association between *INHBC* and serum urate in the EA and AA combined population had a SKAT-O p-value of 2.1e-7 when all putative damaging variants were included and weighted by CADD score. This association was largely driven by the common missense variant (12:57843711:G:A, rs2229357, MAF=0.20, CADD phred score=24.4, single variant association beta=0.08, p=1.8e-7). After excluding rs2229357 from the SKAT-O, the CADD score weighted SKAT-O p-value for *INHBC* was no longer significant (p=0.99). This common missense variant, rs2229357, is in high linkage disequilibrium with the GWAS index variant, rs3741414, in Kottgen *et al*, Nat Genet. 2013 (r²=1, D'=1 in 1000G EUR). The GWAS index SNP rs3741414 is not an exonic variant, rather it is a 3' UTR variant which was not assessed in our study.

Given that common, non-exonic variants can have regulatory effects on neighboring genes and that no functional studies have been performed to support the role of *INHBC* as causally influencing urate levels, we believe it is more conservative to not include statements on the causality of *INHBC* on urate levels in this manuscript. This holds especially true as recent work for diabetes reported that despite the presence of common missense variants in many GWAS loci, compelling evidence to support their causality could only be obtained for about half of the loci, whereas the missense variants represented “false leads” in the other half.²

d. WDR1 and ZNF518B - are these independent from SLC2A9? Conditional analysis should be done.

Response: In published literature pertinent to the Reviewer's question, some studies have reported multiple independent signals in the *SLC2A9* region.^{3, 4} The previous GWAS meta-analysis of serum urate in European ancestry individuals⁵ did not report multiple independent signals in *SLC2A9*, but the criteria to define independence vary across studies. For example, the latter study used a stringent threshold that required <20% change in effect size in conditional analyses to define independence. Thus, while one common variant at *ZNF518B* (rs10939829) achieved genome-wide significance controlling for the index variant at *SLC2A9* ($p=3.7e-10$), the effect size was attenuated by 57.5% and it was therefore not identified as an independent signal.⁵

Similar to the conditional analysis that were conducted for the rare variant in *SLC22A12*, we conducted conditional analysis for all exome-wide significant variants in the *SLC2A9* region reported in **Supplemental Table S14A**, as requested by the Reviewer. In these analyses using data from the European American cohort of the large ARIC study, we controlled for the GWAS index SNP, rs12498742, reported in Kottgen *et al.* 2013. Consistent with the results in Kottgen *et al.*, the association between urate and all these variants were attenuated after controlling for rs12498742 with a reduction in the absolute effect size of 33% to 60%, although rs10939650 retained an exome-wide significant association p-value (conditional $p=2.1e-8$, **Supplemental Table S14B**). In addition to presenting these results as a Supplemental Table, we have added this analysis to the manuscript (**Methods: page 7, paragraph 3; Result: page 13, paragraph 3**).

2. While the association data for rs150255373 with blood pressure are interesting, they are buried in the paper and not specifically mentioned in the main text. Either remove or report in main text.

Response: We have now reported this analysis briefly in the main text (**page 14, paragraph 1**)

3. Line 291, incorrect Supp Figure referred to.

Response: At line 291, we referred to Supplemental Figure S3, which presents all potentially damaging rare variants included in the primary gene-based analysis in *SLC22A12* as a lollipop plot. To clarify this, we have added "lollipop plot" to the text (**page 14 paragraph 1**).

4. Line 338-339. That the four *SLC22A12* variants explain 0.8% of phenotypic variance is remarkable. Could the authors please calculate the phenotypic variance explained by all coding variants listed in Table 15B.

Response: **Table S15B** included all putative damaging coding variants ($n=12$) that met the significance threshold for the single variant interrogation of previously reported urate GWAS loci. To generate the phenotypic variance explained as requested by the Reviewer, we used the formula: phenotypic variance explained = $\beta^2 * 2 * (MAF) * (1-MAF) / \text{phenotypic variance}$. To avoid double counting and thereby inflating the estimate of the explained phenotypic variance, for variants with pairwise $r^2 > 0.2$, we selected the variant with the lower association p-value, resulting in the exclusion of rs2280205 (1000G EUR r^2 with rs2276961: 0.2) and rs11754288 (1000G EUR r^2 with rs1165196: 1.0). The total phenotypic variance explained by the remaining 10 SNPs, including the two low-frequency variants discovered in the study, was 7.3%. This estimate is consistent with the proportion reported in Kottgen *et al.* 2013 for the 26 replicated variants (7%) because the common variants in Table 15B are in LD with the GWAS index SNPs as shown in the conditional analysis (response to 1d) and included variants in loci with the largest effects for urate (*SLC2A9* and *ABCG2*) in Kottgen *et al.* 2013. Therefore, we would not interpret the 7.3% as phenotype variance explained by missense variants. Our study is smaller

than the GWAS in Kottgen *et al.* It is possible that the estimate of phenotypic variance explained may be higher in smaller studies with less heterogeneous populations. In addition, most of these variants have not shown to be causal (see also Response 1c).

5. Throughout be consistent between the use of uric acid / urate. The latter is preferable.

Response: We have taken the advice of the Reviewer and now use “urate” throughout the manuscript when referring to the molecule at physiological pH (e.g. transport studies, residues potentially involved with urate binding, etc), because at physiological pH the dominant form is the disassociated anion urate. However, we refer to uric acid as “C-14 Uric Acid” in the Methods section because the chemical reagent we used is the acid, this reduces confusion for others who want to follow our methods or order similar reagents.

6. Generally, it is preferable to use the word 'genetic variant / variation' rather than mutation.

Response: We have used the term “genetic variant” throughout the manuscript..

7. Lines 389-390. Here it is implied that individuals taking urate-lowering therapy were included in the analysis. This would be far from ideal. Please confirm that this wasn't the case. If it was, then such individuals should be excluded.

Response: We agree with the Reviewer that the inclusion of participants who were on urate-lowering therapy (ULT) in the urate analysis is not ideal. **Supplemental Table 9** reported the available information on gout medication use. As the Reviewer can see, four of the eight studies that contributed to the serum urate analysis did not collect information on ULT. This represents one of the differences between the conduct of research in population-based studies and research among clinical populations ascertained for a specific purpose, in which more detailed information is often available. If we had excluded participants on ULT in the serum urate analyses, studies without ULT information would have to be excluded entirely, resulting in substantially smaller sample size and diminishing statistical power, or the participant inclusion criteria would not have been uniform across studies. We therefore deliberately decided to include participants who were on ULT in our analyses, after confirming that the proportions of participants on ULT were small (1.1% to 2.7% across cohorts with available information). Again, in population-based studies, the proportion of individuals on ULT is expected to be small, as opposed to clinical gout populations. We acknowledged that potential gain-of-function variants might have been missed by including a small proportion of individuals on ULT in the **Discussion section**. In addition, we have now added another sentence to the Discussion contrasting population-based and clinical samples and their complementary value of studying the genetics of urate metabolism and gout (**Discussion section, page 20, paragraph 2**).

8. That the rs12800450 variant originally reported by Tin *et al* was not detected in this study is extraordinary. HapMap predicts an allele frequency of ~1% in people of European and African ancestry. Therefore for this variant to be not present at all does imply an error (technical or analytical) at some point, that may be specific for this variant, or more general. Could the authors please investigate this further.

Response: We agree with the Reviewer that this point requires further investigation. Our previous study in 2011 followed established practices for GWAS: the significant association observed in the discovery sample was replicated in the independent replication sample, and functional examination of the resulting amino acid change were performed. In our 2011 report, we described a significant association between a low-frequency variant in *SLC22A12*, rs12800450, and serum urate among African American individuals. The variant was not directly genotyped in these study samples, but imputed from the HapMap 2 data. Based on the HapMap2 genotypes available from Ensembl, one individual in each of the YRI and CEU

samples carried one copy of this variant. The HapMap2 CEU and YRI sample haplotypes were used as the reference for imputation.⁴ All variants with imputation quality >0.3 were included in the analysis, following common practice for imputation quality cutoff.⁶

One way to further investigate the genotype of rs12800450 in our previous study population would be to directly genotype this variant, for example using a Taqman assay. Unfortunately, we are unable to obtain blood samples from the study participants in our previous report to perform genotyping. Although we cannot currently further investigate this angle through genotyping, it was still important to us to mention that the variant was not detected in the WES data in our current report. This will allow others and ourselves to further investigate the frequency of the variant in future sequencing studies. Importantly, upcoming data from the large, trans-ethnic TOPMed Program will provide an additional opportunity to further investigate the allele frequency of this variant in whole genome sequence data from diverse ancestries. We have added this last point to the Discussion Section (**page 23, paragraph 2**).

The other significant finding in our 2011 report was that the rs12800450 variant, coding an amino acid change G65W, reduced the urate transport function of URAT1. However, we were unable to perform a key experiment because of the lack of a URAT1 antibody. Now, using the same anti-URAT1 [SLC22A12] antibody described in our current manuscript, a critical reagent unavailable in 2011, we have assessed URAT1 protein abundance of the G65W variant. We found that the G65W variant expresses protein at much lower levels than the Wt (**Reviewer Figure R2**), a finding that correlates to predicted trafficking errors from the alterations

to the first extracellular loop and prominent glycosylation sites. These new findings are consistent with experimental data included in our 2011 publication, and provide a clear molecular mechanism for the serum urate lowering effect of the rs12800450 T allele.

9. Figure 1. Panel B appears self-serving. The rare variants were originally selected as beta<0 and beta>0 for inclusion in the plot. Panel C - why not plot the absolute value of beta ?

Response: The effect estimates (betas) in the meta-analysis represents the mean difference among genotypes using the additive genetic model and do not provide information on the full distribution of the outcome by genotype category, which were shown in Panel B. For example, some carriers of variants with negative betas had serum urate levels as high as 10 mg/dL, likely reflecting the influence of environmental and other genetic factors. As suggested by the Reviewer and for the Reviewer's information, **Reviewer Figure R3** shows the distributions of

Figure R3. Distributions of serum urate in ARIC in those with and without putative functional variants in SLC22A12

serum urate among those without any putative damaging variants and those with at least one, i.e. combining both positive and negative betas. We understand the Reviewer's point and have removed Panel B from **Figure 1**.

Reviewer #2 (Remarks to the Author):

The authors conducted an exome-wide association study of kidney function-related traits, using ~20,000 European and African subjects. By applying a set of association analyses targeting not only common variants but also low-frequency or rare alleles. Gene-based test identified rare variant risk on uric acid levels at *SLC2A9* and *SLC22A12* genes. Subsequent analysis validated functional roles of these variants on the traits. This is a well-designed and powered genetic study to assess roles of rare variants on human complex traits, and this reviewer has few comments.

Response: We thank the Reviewer for the positive feedback.

1. While the authors assessed several kidney function-related traits, significant signals were only observed at uric acid. This may imply relatively larger impacts of rare variants on uric acid rather than other traits. Discussions on why it happens may be useful (e.g. selection pressure, functional character of transporter genes, etc).

Response: That significant associations were only observed with serum urate and not with eGFR or UACR could be due to several reasons, or a combination thereof: (1) The effect size of variants for eGFR and UACR may be smaller than those for serum urate. Based on GWAS findings, the serum urate variance explained by the 26 replicated index SNPs was 7% (N~140,000, Kottgen *et al.* 2013). In contrast, 53 eGFR index SNPs explained only 3.2% of the eGFR variance (N~133,000, Pattaro *et al.* 2016).^{5, 7} The smaller effect size for kidney related traits may be due to biology or imprecision in the measurement of eGFR and UACR, while the short-term variability of serum urate was found to be relatively small.^{8, 9, 10} (2) Rare variants of large effects may be under more stringent purifying selection for eGFR and UACR than for urate. This would be consistent with the observation that most adverse outcomes that have been related to urate are associated with higher and not lower urate levels, potentially removing purifying selective pressure on rare urate-lowering variants of large effect. This could give rise to differences in allelic architecture, such that rare variants of large effect are observed only for urate but not other kidney function measures. We have expanded the **Discussion section** devoted to these points (**page 23, paragraph 2**).

2. Variant associations of *SLC2A9* and *SLC22A12* on uric acid has been reported in many populations including east Asians, which should be mentioned.

Response: We now conducted a search for reported coding variants associated with serum urate in *SLC2A9* and *SLC22A12* using the keywords: "human", "mutation", and "urate", combined with "*SLC2A9*" or "*SLC22A12*". We then excluded functional, AKI, or lab method studies, non-English publications, studies reporting known variants, or reviews and identified 21 publications, which are listed below (**Table R1**) and now cited in the supplemental materials of our manuscript due to the reference limit of Nature Communications (≤ 70). Of these 21 studies, 20 reported on the association with hypouricemia. Of the coding variants that were reported in these publications, 14 variants were found in our study (two in *SLC2A9* and 12 in *SLC22A12*). Of the 14 variants, 12 were associated with lower serum urate levels. Overall these results were consistent with previous reports and were included in the manuscript (**Results, page 15, paragraph 3; Supplemental Table S15A**, search details added to the **Supplementary Methods**).

Table R1. Publications reporting the association of coding variants in *SLC2A9* and *SLC22A12* with serum urate.

First author	Year	PMID
Enomoto	2002	12024214
Ichida	2004	14694169
Iwai	2004	15327384
Wakida	2005	15634722
Vázquez-Mellado	2007	17660342
Lam	2008	18760270
Matsuo	2008	19026395
Dinour	2010	19926891
Dinour	2011	21148271
Kawamura	2011	22132964
Tasic	2011	22194875
Dinour	2012	21810765
Li	2013	23043931
Stiburkova	2013	23386035
Jeannin	2014	24397858
Androvitsanea	2015	25966807
Fujita	2016	27780716
Mancikova	2016	26500098
Okabayashi	2016	26971588
Windpessl	2016	27116386
Claverie-Martin	2018	29486147

Reviewer #3 (Remarks to the Author):

Well written manuscript with information clearly presented, particularly in the supplementary material. This manuscript describes exome resequencing studies in up to 15,431 individuals of European and 3,698 individuals of African American descent, which identified multiple low frequency SNPs associated with serum urate and having functional effects. Clarification of several questions would improve evaluation of the data described in this manuscript.

Response: We thank the Reviewer for this positive feedback.

1. The demographics and biochemistry measurements reported for participants in S10A raises questions as to whether the MDRD-4v equation is best suited to estimate GFR in these individuals, particularly older participants; for 5+ years the CKD-EPI equation has been commonly used by many large commercial clinical labs for eGFR reporting. CKD-EPI reduces bias where eGFR>60 and was recommended by KDIGO 2013. The mean eGFRcr in each cohort included in this study is >70; does using different equations to calculate eGFR affect relevant results? Was serum urate levels measured outside an acute flare for relevant participants and were participants taking any urate lowering medication?

Response: We agree that the more recent CKD-EPI equation estimates GFR with higher precision and accuracy than the MDRD equation, especially at higher levels of eGFR. When this project started three years ago (March 2015), other ongoing kidney function GWAS were using the MDRD equation.^{7, 11, 12, 13} We therefore decided to be consistent with these projects in order to facilitate potential comparisons. Theoretically, the lower precision of the MDRD equation may

have reduced our ability to detect associations for eGFR for genes related to kidney function in the higher eGFR range (≥ 60 mL/min/1.73m²). However, other GWAS of eGFR that have used the CKD-EPI equation have reported the same loci identified in our earlier GWAS using the MDRD equation, and common eGFR-associated variants reported in our current study were also found in GWAS using the CKD-EPI equation.^{14, 15} As the rare variants that we were hoping to find in our study were expected to have larger effects than the common variants identified using the MDRD and CKD-EPI equations previously, we believe that the use of the MDRD equation to estimate GFR does not explain that no rare variants of large effect associated with eGFR were identified in our current study. Please see response to Reviewer 2 comment #1 for additional potential reasons for the null results on kidney function.

The point raised by the Reviewer related to gout flares at the time of urate measurement would certainly be of relevance in an ascertained case-control study for gout. We, on the other hand, conducted our studies of serum urate in population-based studies, where urate was measured in blood samples collected at a scheduled study visit. It is not likely that a participant would attend a voluntary study visit that lasted several hours during a gout flare. However, we do not have this information. The proportions of participants who were on urate lowering medications are small and reported in **Supplemental Table S9**. Please see additional discuss on this point in response to Reviewer 1 comment #7.

2. Please include a matrix summarizing the power of this study to identify race specific effects for low frequency variants for the primary phenotypes in the methods. For the aggregated results, was the effect of having specific combinations of SNPs considered? Were any of the proposed-risk rare variants in SLC22A12 and SLC2A9 identified in control populations?

Response: We calculated the minimum detectable effect sizes in unit of SD for single variant analysis for MAF of 0.1%, 1% and 5% using the sample sizes overall and in European and African ancestries for serum urate, eGFR, and UACR separately, assuming an alpha of 7.8e-8 accounting for multiple testing correction and 80% power (**Table R2, Supplemental Table S10**). The effect sizes of variants detected for serum urate in our study were consistent with the power calculations. For example, the minimum detectable effect size for serum urate was 1.09 for a MAF of 0.1% given a sample size of 16189, and the effect size of rs150255373 (MAF 0.1%) was -1.11.

For the tests aggregating putative damaging variants, the aggregation unit was the gene that was linked to the variant based on ANNOVAR annotation. We aggregated variants with MAF < 1% and < 5% as our primary analyses since power is lower for detecting the association of low frequency variants separately given the same sample size. These tests aggregate rare variants within a gene accounting for the covariance between the variants to avoid overestimation of the aggregate association due to correlation between variants. Explicit combination or phase information was not considered. Since the variants were very rare, individuals almost always carried only one of the variants.

To examine whether the two low-frequency urate-associated variants we report in **Table 1** exist in control populations, we queried the ExAC Browser and found both (rs150255373 T allele, frequency 0.04% in non-Finnish Europeans; rs147647315 A allele, frequency 1.7% in African ancestry). However, since we do not know the serum urate values of the individuals in these databases and general population-based samples include individuals with very low urate levels, the definition of “control population” is difficult. Again, this is a difference between population-based research and genetic research among clinical and ascertained populations.

Table R2. Minimum detectable effect sizes based on ancestry-specific sample sizes assuming an alpha of 7.8e-8 and 80% power

Trait	Ethnicity	N	MAF	Minimum detectable effect size in SD
Urate	Overall	16189	0.001	1.09
Urate	Overall	16189	0.01	0.35
Urate	Overall	16189	0.05	0.16
Urate	EA	13363	0.001	1.22
Urate	EA	13363	0.01	0.39
Urate	EA	13363	0.05	0.18
Urate	AA	2826	0.001	2.60
Urate	AA	2826	0.01	0.83
Urate	AA	2826	0.05	0.38
eGFR	Overall	18795	0.001	1.02
eGFR	Overall	18795	0.01	0.32
eGFR	Overall	18795	0.05	0.15
eGFR	EA	15099	0.001	1.13
eGFR	EA	15099	0.01	0.36
eGFR	EA	15099	0.05	0.16
eGFR	AA	3696	0.001	2.31
eGFR	AA	3696	0.01	0.73
eGFR	AA	3696	0.05	0.34
UACR	Overall	12844	0.001	1.22
UACR	Overall	12844	0.01	0.39
UACR	Overall	12844	0.05	0.18
UACR	EA	10431	0.001	1.39
UACR	EA	10431	0.01	0.44
UACR	EA	10431	0.05	0.20
UACR	AA	2413	0.001	2.81
UACR	AA	2413	0.01	0.89
UACR	AA	2413	0.05	0.41

3. Multiple capture methods have been used to enrich the exons – what is covered and how many exons / entire genes are missed based on each approach? Were many miRNA genes covered by the selected capture approaches?

Response: Based on independent research, the exome capture methods used in our cohorts (NimbleGen SeqCap EZ and Agilent SureSelect v4 or v5) covered most of the coding regions (ranged 76% to >90%)^{16, 17} and nearly 80% of all miRNA in miRBase v.15.¹⁸ The coverage differs by exome capture kit. The estimate for the same kit can also differ by the reference gene set, e.g. using RefSeq coding regions only or combining coding regions from multiple databases.^{16, 17} It is therefore difficult to quantify exactly the proportion of covered genes and exons. Since the gap in coverage may have limited our ability to detect genetic association in those regions, we have added this point to the **Discussion section (page 23, paragraph 2)**.

4. The QC filters for original data calling are different between the groups; was this apparent during the analysis e.g. were SNP calls (MAF / MAC rates) consistent between cohorts? What minimum read depth was used for each cohort when calling variants? What was the average number and range of SNPs identified per exome in each cohort? What does ‘swaps fixed’ mean?

Response: We understand the Reviewer's points on the consistency of variant calling across cohorts, a problem that necessarily occurs when data from different studies is combined in order to reach the large sample sizes required for the detection of rare variants. The minimum read depth was 10 across cohorts. Most of the samples (ARIC, CHS, and FHS, ~60% of the overall sample size) were sequencing together with the same QC criteria. Allele frequency and number of SNPs detected across cohorts may not adequately reflect consistency of genotype calls because allele frequency depends on genetic ancestry and population substructure within an ancestry. For example, the Erasmus Rucphen Family (ERF) study is an isolate population.¹⁹

Regarding the number of variants per exome, we now computed these statistics in the ARIC study, which used NimbleGen SeqCap EZ only. In this study, we found an average of 19,468 (min 13,505, max 23,525) variants per exome for European ancestry individuals and 24,311 (min 17,941, max 25,723) for African Americans. These results are consistent with previous reports. For example, Hoischen *et al.* reported a mean of 21,800 variants from four individuals of European ancestry using Agilent SureSelect human exome kit.²⁰ Similarly and very consistent with our numbers, Bamshad *et al.* reported an average of ~20,000 variants in European American samples and ~24,000 variants in African American samples identified through exome sequencing in human populations.²¹ Our cohorts shared summary results for meta-analyses, and the mean numbers of variants per exome from other cohorts are therefore not readily available.

If different filtering criteria across cohorts led to different call sets, these differences were likely independent of the phenotypes and would limit our ability to detect true associations. We have added this point to the **Discussion section (page 23, paragraph 2)**.

"Swaps fixed" referred to checking of sex mismatch, which is not directly related to variant filtering. We have removed this from the **Supplemental Table S1B**.

5. SNPs have been previously reported as associated with [serum urate] based on imputed and directly typed GWAS data; e.g. from >140,000 individuals in which many cohorts used minor allele frequency <1% as part of their QC filter. For cohorts that have both exome sequencing and high density imputed data, would the SNPs (MAF>1<5) have been identified regardless of which approach was used for primary genotyping? i.e. does the high density imputation data for these lower frequency variants equate to the NGS results? From the 16 variants identified with exome-wide significance (line 268), 14 were common – were these all previously identified by larger GWAS for serum urate levels? How many novel SNPs (MAF>1≤5; MAF≤1) were identified by WES that were not available in the 1 KGPv3 reference imputation data.

Response: As the Reviewer points out, the previous urate GWAS with N>140,000 (Kottgen *et al.* 2013) did not report on variants below MAF of 1%. As requested by the Reviewer, for significant variants with MAF between 1% and 5%, we now compared their association with serum urate between genotypes obtained from imputation versus those obtained from exome sequencing in the ARIC study where both types of information were available. The only significant variant of MAF between 1% and 5% for which this data was available was rs147647315 in African Americans. Using 1000G imputed dosages and exome sequencing genotypes among 2,190 participants of the ARIC study, the effects were comparable (imputed dosage A allele beta= -0.66, p=3.1e-6, imputation quality=0.85; exome sequencing A allele beta= -0.71, p=2.0e-7). Based only on these results on rs147647315, it is difficult to conclude whether the somewhat larger effect sizes observed for sequencing than for imputation would lead to differential detection of a variant. More generally, sequencing and imputation have been compared in other large datasets. For example, Extended Figure 9 in the 1000 Genomes Project

publication in Nature 2015 shows that the performance of imputation to detect variants identified by sequencing declines below 5% MAF.²²

As suggested by the Reviewer, we have computed the proportion of variants detected in exome sequencing and not in 1000G phase 3, the proportions were 80% among variants with MAF <1% and 24% among variants with MAF between 1% and 5% in the urate primary single variant meta-analysis.

Of the 14 urate-associated variants in **Supplemental Table S14A**, five were not available in the GUGC results: one was the low frequency variant from the triallelic rs10939650 (C/G, 836 copies), and the other four were common. Since the GUGC GWAS used the HapMap2 haplotypes as the imputation reference panel, which did not contain triallelic SNPs, we excluded rs10939650 C/G from the lookup while rs10939650 C/T (MAF 26%) was available. For the other four common variants, we identified proxies using r^2 in the 1000G CEU population. As the Reviewer can see, all 13 variants were genome-wide significant in the GUGC urate GWAS (**Table R3**).

Table R3. GUGC urate GWAS results for the 13 common urate-associated variants in Supplemental Table S14A.

Variant	Proxy (r^2 , noncoded/ coded alleles)	Gene	EA and AA meta-analysis using genotype from exome sequencing				GUGC	
			Ref. allele	Alt allele	Beta (SD)	P-Value	Beta (mg/dL)	P-Value
rs1260326	--	GCKR	T	C	-0.08	1.4E-12	-0.08	1.3E-40
rs16890979	--	SLC2A9	C	T	-0.30	4.0E-116	-0.38	<1e308
rs13125646	--	SLC2A9	A	G	0.29	9.8E-105	0.36	<1e308
rs10939650	--	SLC2A9	C	T	0.30	4.2E-112	0.36	<1e308
rs10939650	--	SLC2A9	C	G	0.26	5.4E-11	--	--
rs13113918	--	SLC2A9	A	G	0.33	3.1E-129	0.38	<1e308
rs2276961	rs2240720 (1.0, C/T)	SLC2A9	C	T	0.16	3.1E-41	0.19	1.8E-242
rs35782983	rs17251963 (0.91, T/C)	WDR1	G	A	-0.31	2.1E-94	-0.34	<1e308
rs10016022	--	ZNF518B	A	G	0.07	2.0E-08	0.09	3.0E-43
rs66538112	rs10016022 (0.96, G/A)	ZNF518B	C	G	-0.08	1.3E-08	-0.09	3.0E-43
rs2231142	--	ABCG2	G	T	0.22	2.4E-32	0.22	4.4E-116
rs11754288	--	SLC17A4	G	A	-0.08	1.4E-11	-0.09	9.4E-49
rs1165196	--	SLC17A1	G	A	0.08	3.1E-13	0.09	2.1E-57
rs526338	rs11231845 (1.0, G/A)	NRXN2	G	A	-0.07	4.6E-09	-0.06	6.8E-24

6. For the listed SNPs with most evidence of significance based on a single variant test or burden analysis, was significance identified and a consistent direction of effect observed across all cohorts?

Response: The directions of association of the two exome-wide significant variants in **Table 1** were consistent across cohorts: rs150255373 effect: ARIC EA (-1.29), CHS (-0.91), FHS (-1.12), RS (-0.54), CoLaus (-0.97), not detected in ARIC AA, ERF, and CILENTO. For rs147647315, the effects were: ARIC EA (-0.98), ARIC AA (-0.71), CHS (-1.60), FHS (-0.72), not detected in RS, ERF, CoLaus, and CILENTO. This information is added as a footnote to **Table 1 and in Supplemental Table S14A**.

7. It is surprising that a whole genome sequencing study identifying lower frequency variants associated with gout and serum uric acid levels was not referenced in this manuscript; was there any trend towards association in this study for Sulem and colleagues' low-frequency missense variant (c.1580C>G in

ALDH16A1) associated with gout (OR = 3.12, P = 1.5×10^{-16} , at-risk allele frequency = 0.02) and serum uric acid levels (effect = 0.36 P = 4.5×10^{-21} (Nat Genet. 2011 43(11):1127-30. doi: 10.1038/ng.972).

Response: Thank you for raising this interesting point. Consistent with previous report, c1580C>g, (rs150414818 at 49969006, b37) was associated with higher serum urate levels and higher risk of gout in our study (serum urate beta: 0.45 SD, p=0.003, MAF=0.16%, n=13,776; gout OR=5.93, p=0.02, MAF: 0.14%, n=4,959). We have added these results to the manuscript (**page 15, paragraph 3**).

8. Given the traditional increased risk of being male for these phenotypes, was sex-specific meta-analyses considered for all genes? Were reference ranges for lab tests different for males and females? If so, was this considered in the relevant association analyses?

Response: We did not perform sex-specific analysis, because it would have resulted in lower power than that for overall analysis given smaller sample sizes. Since minimum minor allele count filters were applied to variants, sex-stratified analyses would have resulted in a loss of information since many rare variants passed this threshold only in the combined sample. Serum urate was analyzed as a continuous trait, which does not required the use of reference ranges by sex that is commonly applied to define hyperuricemia.

Minor comments:

a) I do not see any data or code availability statements in the manuscript.

Response: The code for meta-analysis is included as Supplemental Material. The meta-analysis results will be posted in dbGaP.

b) It would be helpful in supplemental Figure S1 to include numbers of individuals / snps / genes considered at relevant stages.

Response: We have now added sample sizes, number of variants and genes in the Figure S1 or as footnotes of the figure.

c) Several of the references are incomplete e.g. ref 35 Voorman et al.

Response: Thank you for pointing this out. The references that appeared incomplete were software packages, webpages, or conference proceedings. We have reformatted these references to remove extraneous characters.

d) There are font changes within the supplementary tables e.g. S1B cell E4 vs E5 is Arial -> Calibri

Response: Thank you for pointing this out. We have standardized the font to Arial.

e) Is it possible to include the direction of effect in each cohort for SNPS reported from the meta-analyses in supplementary material?

Response: It is certainly possible to include the direction of effect in each cohort. We have included the beta in each cohort for the variants reported in **Table 1** as a footnote and as separate columns in **Supplemental Table S14A** and **S15A**.

References

1. Pautsch A, *et al.* Crystal structure of glucokinase regulatory protein. *Biochemistry* **52**, 3523-3531 (2013).
2. Mahajan A, *et al.* Refining the accuracy of validated target identification through coding variant fine-mapping in type 2 diabetes. *Nat Genet* **50**, 559-571 (2018).
3. Ware EB, *et al.* SLC2A9 Genotype Is Associated with SLC2A9 Gene Expression and Urinary Uric Acid Concentration. *PLoS One* **10**, e0128593 (2015).
4. Tin A, *et al.* Genome-wide association study for serum urate concentrations and gout among African Americans identifies genomic risk loci and a novel URAT1 loss-of-function allele. *Hum Mol Genet* **20**, 4056-4068 (2011).
5. Kottgen A, *et al.* Genome-wide association analyses identify 18 new loci associated with serum urate concentrations. *Nat Genet* **45**, 145-154 (2013).
6. Li Y, Willer CJ, Ding J, Scheet P, Abecasis GR. MaCH: using sequence and genotype data to estimate haplotypes and unobserved genotypes. *Genet Epidemiol* **34**, 816-834 (2010).
7. Pattaro C, *et al.* Genetic associations at 53 loci highlight cell types and biological pathways relevant for kidney function. *Nat Commun* **7**, 10023 (2016).
8. Naresh CN, Hayen A, Weening A, Craig JC, Chadban SJ. Day-to-day variability in spot urine albumin-creatinine ratio. *Am J Kidney Dis* **62**, 1095-1101 (2013).
9. Levey AS, Bosch JP, Lewis JB, Greene T, Rogers N, Roth D. A more accurate method to estimate glomerular filtration rate from serum creatinine: a new prediction equation. Modification of Diet in Renal Disease Study Group. *Ann Intern Med* **130**, 461-470 (1999).
10. Rubin R, Plag J, Arthur R, Clark B, RH R. Serum Uric Acid Levels: Diurnal and Hebdomadal Variability in Normoactive Subjects. *JAMA* **208**, 3 (1969).
11. Gorski M, *et al.* 1000 Genomes-based meta-analysis identifies 10 novel loci for kidney function. *Sci Rep* **7**, 45040 (2017).
12. Mahajan A, *et al.* Trans-ethnic Fine Mapping Highlights Kidney-Function Genes Linked to Salt Sensitivity. *Am J Hum Genet* **99**, 636-646 (2016).
13. Li M, *et al.* SOS2 and ACP1 Loci Identified through Large-Scale Exome Chip Analysis Regulate Kidney Development and Function. *J Am Soc Nephrol* **28**, 981-994 (2017).
14. Okada Y, *et al.* Meta-analysis identifies multiple loci associated with kidney function-related traits in east Asian populations. *Nat Genet* **44**, 904-909 (2012).

15. Kanai M, *et al.* Genetic analysis of quantitative traits in the Japanese population links cell types to complex human diseases. *Nat Genet* **50**, 390-400 (2018).
16. Meienberg J, *et al.* New insights into the performance of human whole-exome capture platforms. *Nucleic Acids Res* **43**, e76 (2015).
17. Asan, *et al.* Comprehensive comparison of three commercial human whole-exome capture platforms. *Genome Biol* **12**, R95 (2011).
18. Sulonen AM, *et al.* Comparison of solution-based exome capture methods for next generation sequencing. *Genome Biol* **12**, R94 (2011).
19. Pattaro C, *et al.* A meta-analysis of genome-wide data from five European isolates reveals an association of COL22A1, SYT1, and GABRR2 with serum creatinine level. *BMC Med Genet* **11**, 41 (2010).
20. Hoischen A, *et al.* De novo mutations of SETBP1 cause Schinzel-Giedion syndrome. *Nat Genet* **42**, 483-485 (2010).
21. Bamshad MJ, *et al.* Exome sequencing as a tool for Mendelian disease gene discovery. *Nat Rev Genet* **12**, 745-755 (2011).
22. Auton A, *et al.* A global reference for human genetic variation. *Nature* **526**, 68-74 (2015).

Reviewer #1 (Remarks to the Author):

Thanks for addressing my comments and concerns.

My only suggestions are:

1. I accept the logic for not following up on INHBC. Could the authors please include a sentence or two that reflects this logic - i.e. after outlining the LD of the INHBC missense variant with the main 2013 GWAS variant that taking a cautious approach to following up on missense variants and inferring causality is a prudent approach to take.

Response: We agree that this is a useful addition We have added the linkage disequilibrium information (r^2 and D') between the GWAS index SNPs and the SNPs from exome sequencing in **Supplemental Table 15C**, noted the LD information in the **Results** section on **page 16, paragraph 2**. In addition, in this manuscript, we focused our follow-up on low frequency and rare variants. We added the need for functional studies to follow up on common missense variants in the **Discussion** section (**page 24, paragraph 1**).

2. If possible please mention the GCKR structural modelling in the text (data wouldn't necessarily need to be shown).

Response: As outlined in the response above, we focused our follow-up investigation on low frequency or rare missense variants, instead of common missense variants in GWAS regions. Since the GCKR variant (P446L) is a common missense, we find it difficult to find an appropriate place to introduce the modeling results into the manuscript without confusing the readers. We had generated this figure for the Reviewer's information only to address his/her previous question related to this variant. We hope that this is agreeable to the Reviewer and the Editors and would rather not include the GCKR structural modeling into the manuscript.

Reviewer #3 (Remarks to the Author):

Thank you for clearly responding to all issues raised during this review and providing additional material.

Response: We appreciate the Reviewer's positive assessment of our revised work.

Reviewer #4 (Remarks to the Author):

An important part of the manuscript are (1) the expression levels of the wt URAT1 transporter and its mutants in HEK cells, and the C-14 uptake rates determined in oocytes. There are several issues that need to be clarified.

1. Does the Western blot shown in Fig. 2B (upper) represent a single gel or is it composed from different gels?

Response: The line in the images denotes the separation between two separate blots, per *Nature* "Image Integrity and standards" for blot presentation. We have now added this information to the figure footnote (**page 36**).

2. Although two rather diffuse bands (one more than the other) can be recognized, there is also smear above the upper band. Does this smear indicate background or specific labeling? The authors need to show the entire gel and not just a cut-out portion.

Response: We agree that it is important to confirm specificity of the antibody and have therefore carried out antibody validation experiments, presented in an updated **Supplemental Figure S4**. In part A of the Figure, we show the entire Western blot and demonstrate there is no signal detected in lysate from untransfected cells (neither COS7 nor HEK 293, two left lanes). The smear and multiple bands can

only be seen in cells transfected with wild-type (wt) or mutant URAT1 (two right lanes). All bands on the blots therefore indicate specific labeling. In addition, in **Supplemental Figure S4C**, we have now included the entire oocyte Western blot showing again the specificity of the antibody and the existence of specific URAT1 bands at higher molecular weights.

3. Which portion of a lane has been used for quantification of URAT1 protein? How large was the mask?

Response: We have quantified the monomer band (both the glycosylated and non-glycosylated forms). At the Reviewer's suggestion, we have now included an example of the mask (area of densitometry analysis) used for quantifying the monomer bands (both the glycosylated and unglycosylated forms) in **Supplemental Figure S4A**.

4. Multiple diffuse bands result from complex glycosylation patterns and are inherently difficult to quantify. The authors should execute quantification on samples that have treated with PNGase; the smear collapses into a single distinct band that can be easily quantified. At least, the authors should show that quantification before and after PNGase treatment is similar.

Response: As outlined above, in **Figure 2**, we have quantified the glycosylated and non-glycosylated forms of the monomer together. We agree with the Reviewer that the question of glycosylation is relevant and interesting, and have addressed this using both PNGase and ENDOH treatment, included as **Supplemental Figure S5**. At the Reviewer's request, we have closely inspected the three forms of the monomer band, as illustrated by our EndoH and PNGase experiments: non-glycosylated, ER glycosylated (PNGase and EndoH sensitive) and Golgi glycosylated (PNGase sensitive). We have separately quantified the mature band (Golgi glycosylation) and the immature (either no or ER only glycosylation) bands, as well as the ratio of the two. This new analysis is now presented in **Supplemental Figure S5B**. The results support our previous conclusions, because the levels of the mature protein are very similar to our aggregate quantification in **Figure 2**. However from a careful comparison of the ratio of mature to immature protein, we can now more objectively state that the R325W, R405C, and T467M variants lead to an alteration in processing, as reflected by a significantly altered ratio of mature to immature protein as compared to the wild-type. We thank the Reviewer for this useful suggestion, and have included a summary of these findings on **page 17, paragraph 1**.

5. Is there a reason why two lanes for the same mutant or the wt are shown in Fig. 2B?

Response: Each lane is a separate experiment. The data is presented to demonstrate the consistency of the results, in an effort to provide as much raw data as possible for the reader to draw their own conclusions. For the same reason we have included all the actual data points used to determine the mean and SEM for each of our summary data bar graphs. We have now added that each lane is a separate experiment to the figure footnote (**page 36**).

6. Several concerns arise from Fig. 2C. The uptake of C-14 urate (upper) was normalized by Western blots (lower). For the quantification by Western blots, oocytes have been treated with a detergent to solubilize membrane-spanning SLC22A12. Membrane targeting of proteins is a complex process involving several subcellular organelles like the ER or Golgi networks. It is well known that, in heterologous expression, only a minor fraction of the membrane protein makes it to the plasma membrane. Variable protein fractions get stuck to membranes of subcellular organelles along the targeting pathways. However, for normalization of transport only protein residing in the plasma membrane is relevant. Furthermore, most membrane proteins cannot be quantitatively extracted. The efficacy of solubilization – apart from the type of detergent used – depends on detergent concentration (relative to the CMC), protein concentration, and sample volume during solubilization. The authors must test how much of the protein (for comparable sample sizes) remains in the pellet. Furthermore, solubilized protein and protein in the pellet must

arithmetically add up to total protein. Glycosylation often is involved in Protein targeting; and, glycosylation patterns depend on the host cell.

Response: We have carefully considered these points raised by the Reviewer. We agree that Western blots are a semi-quantitative method to determine protein amount, and that not all of the protein will reside in the plasma membrane. In addition, as the Reviewer correctly points out, glycosylation patterns depend on the host cell. Thus, the only way to study protein abundance of the variants detected in our human studies conclusively would be to determine protein abundance in human kidney biopsies from carriers of the reference and alternative alleles at each evaluated variant. Unfortunately, such biopsy or other tissue samples are not available from our population-based studies.

Our original presentation of normalized data was motivated by an attempt to focus on transport function and the fact that our oocyte protein extraction protocol (unlike mammalian cell lysis protocols) includes a differential centrifugation step that removes the egg yolk and most of the yolk proteins¹ (now clarified in the **Methods on page 11 paragraph 1**). This serves to significantly enrich the final protein extraction for plasma membrane proteins. However, after more consideration and an abundance of caution, we now agree with the Reviewer that, given the limitations of Western blots and the use of a model system, the presentation of the normalized data in the main manuscript may introduce additional questions. In response to the Reviewer's comment, we have therefore chosen to revise the **main Figure 2, panel C**, to now display non-normalized transport data. Based on the non-normalized data, measured after injecting the same amount of mRNA of each construct, the inference regarding the variants R325W, R405C, and T467M remains unchanged in that they represent loss of transport function mutations. This is supported by the human evidence (lower serum urate levels), which gave rise to the identification of these variants in the first place. We have adapted the corresponding parts of the main manuscript to reflect this change (**Results, page 18 paragraph 1; page 36 Figure 2C footnote**).

The non-normalized transport rates of the K536T variant are not significantly different than those of the wt URAT1. Curiously, however, the amount of protein as quantified by Western blot was much lower than that of wt or any of the other tested variants, and the human carriers of the variant display higher rather than lower urate levels. Thus, when normalized for protein expression of the respective variant, the normalized transport rates of K536T are higher than those of the wild-type. This may suggest that this variant, if present in the plasma membrane at comparable levels, may show higher transport rates and provide a potential explanation for the human phenotype. The complex phenomenon of higher transport function but reduced protein expression in conjunction with the C-terminal location of the variant points to a role in post-translational regulation via pathways that may be tissue or cell-type specific, making objective determinations using our model systems inconclusive. However, we still believe that this information, previously presented in the main figure, may be of interest to the readers and therefore now present the urate uptake information, normalized to oocyte protein expression, in a new **Supplemental Figure S6B**. Because of the semi-quantitative measurement of protein abundance and the uncertainty about expression levels in human target tissues, we have carefully revised the manuscript throughout. We have discussed the limitations of quantifying protein abundance in our setting (**Results, page 18, paragraph 1**), adapted the manuscript to show and discuss the findings based on the non-normalized data (**Results, page 18, paragraph 1**), and de-emphasized the interpretation of the K536T variant (**abstract; Discussion, page 20, paragraph 1; page 22, paragraph 1; page 24, paragraph 2**).

We thank the Reviewer for bringing to our attention that the previous presentation may have led to over-interpretation of K536T. The conclusions for all other tested variants remain unchanged.

Thus, how relevant are the results from HEK cells for gaining insight into the human disease mechanism?

Response: We wanted to evaluate the potential deleterious nature of the *SLC22A12* mutations not just on transport but also on abundance and trafficking, and chose HEK cells because they are mammalian/human cells and thus better recapitulate what might happen endogenously in humans than would *Xenopus* oocytes. As outlined above, we used these cells as models because the experiments could not be done *in vivo* or using target tissues from the variant carriers and control persons. We concede and understand the limits of the HEK293 model and have carefully constructed our conclusions to allow for the variation between model system and the actual system of interest (**Results, page 18, paragraph 1; Discussion, page 22, paragraph 1**).

On a final note, how large compared to the specific C-14 urate flux was the flux of H₂O-injected.

Response: We agree that this is relevant information, which we have now included in the figure footnote (**Figure 2 legend, page 36**).

7. The Western blots of URAT1 expressed in oocytes look different than those from HEK cells, only a single band is observed. Any explanation?

Response: This can be explained by differences in glycosylation of proteins in *Xenopus* oocytes from those in mammalian cells (see point 6 above). We have now added this observation to the figure legend for **Figure 2 (page 36)**.

8. In the summary figure 2D (lower) the uptake rates of mutant K536T are up by about 20% to 30%, whereas in the upper panel the rate is almost 3.5-fold higher in K536T compared to wt. This is a roughly tenfold difference. This discrepancy casts doubts about the reproducibility or variability of data.

Response: In the previous version of the manuscript, we had presented expression-normalized data from two separate sets of experiments in the two parts of figure panel D. We agree that these contained variability at the urate concentration of 300 μ M and have investigated this further. To account for the full variability of the data, we have now considered data from the two separate sets of experiments together when possible (at the 300 μ M urate concentration point; both **new Figure 2C** and **new Supplemental Figure S6B**). This allows a visualization of the variability of all of the data from multiple experiments and is also now considered in our statistical evaluation of the non-normalized and the normalized data. This accounting for the full variability of the data does not alter the statistical outcomes for the normalized presentation in the **new Supplemental Figure S6B** as compared to the previous **Figure 2D**.

Based on the newly included, non-normalized urate uptake data of K536T (**new Figure 2C**), it becomes however apparent that some of the previous variability could be attributed to the normalization by variable protein expression levels and not to variable transport. We would therefore again like to thank the Reviewer for raising shortcomings of presenting normalized data, and for helping us improve the manuscript.

9. Homology modeling of putative urate-binding sites based on a 3D structure of Glut5 (a fructose transporter) or XyE (a bacterial homologue) has been previously published. The results were completely different (see Cong et al. 2017) and Clemençon et al. 2014), casting doubts about the reliability of this approach. In the absence of a 3D structure of an urate transporter such modeling attempts remain guess work.

Response: We certainly agree that homology modeling has limitations, but feel confident that in the case of GLUT9 it can provide useful insights for a number of reasons. Firstly, there is high homology between the template rat SLC2A5 and human SLC2A9 (45% homology with a TM-align score of 0.882, highest of all structures in the PDB library). Secondly, the use of the rat SLC2A5 template published in *Nature Communications* previously (Long et al 2017)² includes a convincing model of the fructose and glucose substrate binding pockets. In fact, when compared to the modeled GLUT9 substrate-binding pocket proposed by Clemenccon et al (2014)³ using the bacterial Xy1E template, there is considerable overlap in the key residues. This may not be immediately apparent from the literature, because Clemenccon *et al* used the long isoform (NP_064425.2 or SLC2A9a) and Long *et al* used the short isoform (NP_001001290.1 or SLC2A9b) amino acid numbering. When translating the amino acid positions of the Clemenccon model to the version used by Long and colleagues, six of the nine described key residues are common between the two models (L182/153; A206/177; Q328/299; N333/304; W459/430; and N462/433). This supports a very strong conservation of the key binding pocket residues. We thank the Reviewer for bringing the manuscript by Clemenccon *et al* to our attention, and have referenced this work in the **Discussion (page 23, paragraph 1)**.

References

1. Woodward OM, *et al*. Identification of a polycystin-1 cleavage product, P100, that regulates store operated Ca entry through interactions with STIM1. *PLoS One* **5**, e12305 (2010).
2. Long W, *et al*. Identification of Key Residues for Urate Specific Transport in Human Glucose Transporter 9 (hSLC2A9). *Sci Rep* **7**, 41167 (2017).
3. Clemenccon B, *et al*. Expression, purification, and structural insights for the human uric acid transporter, GLUT9, using the *Xenopus laevis* oocytes system. *PLoS One* **9**, e108852 (2014).